



# Ship- and aircraft-based XCH₄ over oceans as new tool for satellite validation

Astrid Müller[1], Hiroshi Tanimoto[1], Takafumi Sugita[1], Prabir K. Patra[2,3], Shin-ichiro Nakaoka[1], Toshinobu Machida[1], Isamu Morino[1], André Butz[4,5], Kei Shiomi[6]

[1]National Institute for Environmental Studies, Tsukuba, Japan
[2]Japan Agency for Marine-Earth Science and Technology, Yokohama, Japan
[3]Research Institute for Humanity and Nature, Kyoto, Japan
[4]Institute of Environmental Physics, University of Heidelberg, Heidelberg, Germany
[5]Heidelberg Center for the Environment, University of Heidelberg, Heidelberg, Germany
[6]Earth Observation Research Center (EORC), Japan Aerospace Exploration Agency (JAXA), Tsukuba, Japan

*Correspondence to*: Hiroshi Tanimoto (tanimoto@nies.go.jp), Astrid Müller (mueller.astrid@nies.go.jp)

**Abstract.** Satellite based estimations of dry-air column-average mixing ratios of methane (XCH₄) contribute to a better understanding of changes in CH₄ emission sources and variations in its atmospheric growth rates. High accuracy of the satellite measurements is required, and therefore, extensive validation is performed, mainly against the Total Carbon Column Observing Network (TCCON). However, validation opportunities at open ocean areas outside the coastal regions are sparse. We propose a new approach to assess the accuracy of satellite derived XCH₄ trends and variations. We combine various ship and aircraft observations with the help of atmospheric chemistry models, mainly used for the stratospheric column, to derive observation-based XCH₄ (obs. XCH₄). Based on our previously developed approach for the application to XCO₂, we investigated 3 different advancements from a simple to more elaborate approaches (approach 1, 2, and 3) to account for the higher tropospheric and stratospheric variability of CH₄ as compared to CO₂. Between 2014–2018, at 20–40° N of the western Pacific, we discuss the uncertainties of the approaches and the derived obs. XCH₄ within 10° by 20° latitude–longitude boxes. Uncertainties were 22 ppb for approach 1, and 17 ppb for approach 2 and 3. We analysed the consistency with the nearest TCCON stations and found agreement of approach 3 with Saga of 1 ± 12 ppb, and −1 ± 11 ppb with Tsukuba for the northern and southern latitude box, respectively. Furthermore, we discuss the impact of the modelled stratospheric column on the derived obs. XCH₄ by applying 3 different models in our approaches. Depending on the models, the difference can be more than 0.5 %, showing the importance for the appropriate choice. We show that our obs. XCH₄ dataset accurately captures seasonal variations of CH₄ over the ocean. Using different retrievals of the Greenhouse gases Observing Satellite (GOSAT) from the National Institute for Environmental Studies (NIES), the RemoTeC full-physics retrieval operated at the Netherlands Institute for Space Research (SRON), and the full-physics retrieval of the University of Leicester (UoL-OCFP), we demonstrate the applicability of the dataset for satellite evaluation. The comparison with results of approach 3 revealed that NIES showed a difference of −0.04 ± 13 ppb and strong scatter at 20–30° N, while RemoTeC



and OCFP have rather systematic negative bias of $-12.1 \pm 8.1$ ppb and $-10.3 \pm 9.6$ ppb. Our new approach to derive $XCH_4$ reference datasets over the ocean can contribute to the validation of existing and upcoming satellite missions in future.

## 1 Introduction

Methane ($CH_4$) is one of the most important anthropogenic greenhouse gases (GHG) in the atmosphere besides carbon dioxide ($CO_2$). Since the pre-industrial reference year of 1750, the annual average surface dry-air mole fraction of $CH_4$ has more than doubled from 729 parts per billion (ppb) to 1866 ppb in 2019 (Canadell et al., 2021). The global warming potential (GWP) over a 100-year period is 28–36 times that of $CO_2$ (Forster et al., 2007). It is estimated that $CH_4$ contributed with 0.5°C to the recent global warming between 2010–2019 relative to 1850–1900 (IPCC, 2021). Compared to $CO_2$, the global atmospheric lifetime of 9.1 years is short (Szopa et al., 2021). Consequently, a reduction in $CH_4$ emission is expected to lead to a quick decrease of global $CH_4$ concentrations and therefore, to short-term mitigation of global warming (Saunois et al., 2020; Shindell et al., 2012).

The wide variation of the mean growth rate of $CH_4$ in the past 3 decades and its rapid rise in the recent years are poorly understood (Canadell et al., 2021; Nisbet et al., 2019; Zhang et al., 2022). While the renewed increase of $CH_4$ was primarily attributed to anthropogenic activities (Zhang et al., 2022), the specific increase in 2020 could be related to lower methane sinks as a consequence of the COVID-19 lockdown and higher wetland emissions (Peng et al., 2022; Stevenson et al., 2022). However, high uncertainties in the processes affecting $CH_4$ sources and sinks remain (e.g., Dlugokencky et al., 2009; Patra et al., 2016; Saunois et al., 2020). With about 90%, the oxidation with OH radicals is the major $CH_4$ sink. It occurs mostly in the troposphere, through which $CH_4$ contributes to the production of tropospheric ozone ($O_3$) (Myhre et al., 2013; Saunois et al., 2020; Kirschke et al., 2013). A smaller part of $CH_4$ is removed by OH oxidation in the stratosphere, where $CH_4$ contributes to the production of stratospheric water vapor (Myhre et al., 2013; Kirschke et al., 2013). An important uncertainty factor in estimating the strength of $CH_4$ sinks is the distribution and variability of OH radicals (Patra et al., 2016; Zhao et al., 2019).

Precise surface and aircraft $CH_4$ in situ measurements are conducted by global networks such as the Cooperative Air Sampling Network of the National Oceanic and Atmospheric Administration Earth System Research Laboratory (NOAA ESRL) (Dlugokencky et al., 2009) and aircraft campaigns such as the HIAPER Pole-to-Pole Observations (HIPPO) campaign (Wofsy, 2011). However, the spatial and temporal coverage is sparse, and vertical coverage is mostly limited to the troposphere. Satellite observations provide global coverage of the column-averaged dry-air mixing ratios of $CH_4$ (denoted $XCH_4$). To obtain information on $CH_4$ sources and sinks, satellite instruments need to be sensitive to variations at near-surface $CH_4$ concentration (Buchwitz et al., 2017). This was given for observations by the Scanning Imaging Absorption Spectrometer for Atmospheric Cartography (SCIAMACHY) on the Environmental Satellite (ENVISAT) (Bovensmann et al., 1999; Frankenberg et al., 2005, Schneising et al., 2011, completed mission 2002–2012), the Thermal And Near infrared Sensor for carbon Observations-Fourier Transform Spectrometer (TANSO-FTS) on-board the Greenhouse gases Observing



Satellite (GOSAT, launched in 2009, Kuze et al., 2009; Yoshida et al., 2011), the Tropospheric Monitoring Instrument (TROPOMI) on board the Sentinel 5 Precursor satellite (launched 2017, Veefkind et al., 2012; Lorente et al., 2021), TANSO-2 on-bord GOSAT-2 (launched in 2018, Suto et al., 2021; Yoshida et al., 2023) or the scheduled GOSAT-GW mission (to be launched 2024, https://gosat-gw.nies.go.jp/en/). These instruments collect spectra of near-infrared (NIR) and shortwave-infrared (SWIR) solar radiation reflected from the Earth's surface, covering the relevant absorption bands of $CO_2$,
$CH_4$ and $O_2$. From these spectra, $XCH_4$ can be derived (e.g., Yoshida et al., 2011, 2013).

Typical variations of $XCH_4$ that relate to sources at the surface are on the order of a few percent at most. Therefore, to be useful for estimating surface fluxes, satellite measurements of $XCH_4$ require high precision, and low random and systematic errors (Buchwitz et al., 2020; Meirink et al., 2006). To achieve these requirements, extensive validation of satellite $XCH_4$ has been performed, mainly against data of the land-based Total Carbon Column Observing Network (TCCON) (Wunch et al.,
2011), which is a network of sun-viewing ground based Fourier transform infrared (FTIR) spectrometers.

70% of the Earth surface is covered by oceans. The marine atmosphere is often influenced by the outflow of continental $CH_4$ emissions, and it is thought that at least half of the $CH_4$ oxidation occurs over oceans (Travis et al., 2020). Satellite retrievals over the oceans, however, have undergone few evaluations since validation opportunities are sparse. They are mostly limited to TCCON sites on islands and the coast, or to episodic measurement campaigns like those of the HIPPO airborne campaign
(Wofsy, 2011) or of individual ship deployments (Klappenbach et al., 2015; Knapp et al., 2021). Continuous reference data of open ocean areas outside the coastal regions remain scarce.

We propose a new approach to assess the accuracy of satellite derived $XCH_4$ trends and variations over open ocean regions by combining commercial ship and various aircraft observations with the help of atmospheric chemistry models. This approach was successfully applied to the evaluation of satellite $XCO_2$ previously (Müller et al., 2021). In contrast to $CO_2$,
$CH_4$ shows higher variability due to its complex interactions between sources and sinks in the troposphere, and additionally, through the stratosphere-troposphere exchange and its stratospheric sinks. To account for this variability, we present the advancement of our previously developed approach and discuss its uncertainties, challenges, and the potential for the continuous validation of satellite observations over oceans in future.

## 2 Observational and model data

### 2.1 Aircraft

As part of Japan's Comprehensive Observation Network for Trace gases by Airliner, CONTRAIL, air samples of $CH_4$ are collected by the Automatic air Sampling Equipment (ASE) and Manual air Sampling Equipment (MSE) about twice a month between Japan, Hawaii, and Australia since 2005. From mid-2017, no data are collected over the western Pacific due to a change of the aircraft type. Within the next 2 years, the resumption of aircraft observations is expected. In cooperation with
Japan Airlines (JAL), the ASE is installed in the cargo compartment on Boeing 747-400 and 777-200ER aircrafts (Machida et al., 2008; Matsueda et al., 2008). Details of the ASE are described elsewhere (Machida et al., 2008; Matsueda et al.,





2008). During one flight, 12 samples are collected at the cruising altitude of about 9–12 km by using the air-conditioning system of the aircraft. The trace gas concentrations were measured at the National Institute for Environmental Studies (NIES), Tsukuba, Japan. The air samples were dried by passing through a glass trap cooled to −80°C (Machida et al., 2008).

The $CH_4$ dry-air mixing ratio of each air sample was determined against the NIES-94 $CH_4$ scale, which is traceable to the standard gas scale of the World Meteorological Organization (WMO) (Dlugokencky, 2005), by using a gas chromatograph equipped with a flame ionization detector (GC-FID; Agilent Technologies, HP-5890 and 7890) (Machida et al., 2008). The analytical precision for repetitive measurements is 1.7 ppb.

Measurements with the MSE are conducted when the ASE cannot be operated. Sample air is taken from the air outlet nozzle

in the cockpit using a manual diaphragm pump. The sampling method is similar to that used during aircraft observations by the Japan Meteorological Agency (JMA) (Tsuboi et al., 2013; Niwa et al., 2014). Only ASE and MSE data which were obtained below the tropopause height during the cruising part of the flight at around 11 km altitude (~200 hPa) are used. We used the blended tropopause pressure (TROPPB) to define the tropopause height, which is explained in detail in **section 2.3**. Data of the lower stratosphere were only occasionally obtained and screened out.

Air samples of the mid-troposphere at about 6 km altitude (~450 hPa) were collected by a cargo aircraft C-130H between Kanagawa (35°27' N, 139°27' E) Prefecture near Tokyo and Minamitorishima (MNM) (24°17' N, 153°59' E) about 2000 km southeast of Tokyo. The observations were conducted by JMA in cooperation with the Japan Ministry of Defense about twice a month, either by direct flights or via Iwo Jima (24°47' N, 141°19'), about 1000 km south of Tokyo. Air samples from the air-conditioning system were collected and analyzed at the JMA using a cavity ring-down spectroscopy (CRDS) analyzer

(Picarro Inc., Santa Clara, CA, USA, G2301) (Saito, 2022). The concentrations of $CH_4$ are determined by the JMA standard gases that are traceable to the WMO standard scales. The reproducibility of $CH_4$ concentration of different flasks has a precision of ±0.68 ppb (Tsuboi et al., 2013).

## 2.2 Ship

Commercial cargo Ships of Opportunity (SOOP) have been collecting air samples since 2001 between Japan and North

America, since 2005 between Japan and Australia and New Zealand, and since 2007, between Japan and Southeast Asia. In this study, we used $CH_4$ observations by the cargo ship Trans Future 5 (TF5, Toyofuji Shipping Co., Ltd.), which sails between Japan, Australia, and New Zealand. Each round trip takes about 5 weeks (Terao et al., 2011). Concentrations of $CH_4$ were continuously measured using CRDS analyzer (Picarro, models EnviroSense 3000i and G1202). In parallel, concentrations of $CO_2$ and $O_3$ were measured. The same instrumentation and analysis methodology was used and described

in detail in Nara et al. (2014). In short, the air intake was placed at the bow on the top of the bridge at about 28 m above sea level, 163 m away from the smokestack at the stern (Terao et al., 2011). Exhaust contaminated samples were rejected when the dry-air mole fractions of $CO_2$ and $O_3$ showed an abrupt increase and decrease, respectively. The analytical precision for 1-min measurements was 0.5 ppb. Calibration with three standard gases was performed for 30 min (10 min for the respective gas) once every two days. The standard gases were calibrated against the NIES-94 $CH_4$ scale.



In addition, atmospheric $CH_4$ data collected by the research vessel Ryofu Maru (RYF, operated by JMA) at the Pacific Ocean were used (Enyo and Kadono, 2021). The intake for air samples was about 8 meters above the sea surface. Air samples were dried and the mole fraction of $CH_4$ was determined by gas chromatography (SHIMADZU, GC-8A). After 2016, data were collected using off-axis integrated cavity output spectroscopy (Los Gatos Research, GGA-30r). Calibration with 3 standard gases was performed every hour, and every 12 hours after 2016.

## 135    2.3 Models

The Model for Interdisciplinary Research On Climate Earth System, version 4.0 (MIROC4) -based Atmospheric Chemistry Transport model (ACTM) has a horizontal resolution of triangular 42 truncation (T42) which corresponds to approximately 2.8° longitude by 2.8° latitude. Details of the MIROC4-ACTM are described in Patra et al. (2018). The MIROC4-ACTM uses 67 vertical layers between the Earth's surface and 0.0128 hPa. Hybrid vertical coordinates are used to resolve gravity wave propagation in the stratosphere, where at least 30 model layers reside. The ACTMs are nudged with the Japanese 55-year Reanalysis data (JRA-55; Kobayashi et al., 2015) for horizontal winds and temperature at Newtonian relaxation times of 1-hour and 5-hours, respectively. A high accuracy of the MIROC4-ACTM is indicated by the agreement of simulated and observed "age of air", and the inter-hemispheric gradient of $SF_6$ (Patra et al., 2018).

The Copernicus Atmosphere Monitoring Service (CAMS), operated by the European Centre for Medium-Range Weather Forecasts (ECMWF), provides global greenhouse gas reanalysis (EGG4) data. The CAMS reanalysis dataset assimilates satellite observations of atmospheric trace gases and global emission datasets. The horizontal resolution at a spectral truncation of T255 corresponds to a 0.7° × 0.7° (longitude–latitude) grid. The vertical model resolution consists of 60 hybrid sigma-pressure levels which are interpolated to 25 pressure levels between 1000 hPa and 1 hPa, with about 12 levels in the stratosphere (Inness et al., 2019). In this study, we used EGG4 $CH_4$ data with monthly average fields, version v20r2. Furthermore, we used the CAMS global inversion-optimized greenhouse gas fluxes and concentrations dataset (CAMSinv) v20r1 which accounts for chemical loss in the troposphere and stratosphere. The inversion-optimized dataset has a horizontal resolution of a 2° × 3° (longitude–latitude) grid, and 34 pressure levels between 1001 hPa and 0.5 hPa (Segers and Steinke, 2022). We choose datasets which assimilate NOAA surface observations, but not GOSAT observations.

Furthermore, we extracted data of the TROPPB, which is defined as a combination of a thermal tropopause- and dynamic tropopause pressure (Wilcox et al., 2012). The TROPPB data are extracted from GEOS-FP IT (Goddard Earth Observing System-Forward Processing for Instrument Teams) meteorology data using the python suite "ginput" version 1.0.6 (Laughner et al., 2022). At 10° × 20° latitude–longitude boxes (**section 3.1**), the TROPPB was calculated daily every 3 hours for the center and the 4 corner locations, and was then monthly averaged.

## 2.4 Satellite

Japan's GOSAT launched in 2009, was developed to characterize the variability of the atmospheric $CO_2$ and $CH_4$ fractions at regional scales over the globe. The TANSO-FTS instrument on board GOSAT measures the reflected sunlight in three SWIR





channels: centered at 0.764 µm (Band 1), at 1.61 µm (Band 2), and at 2.06 µm (Band 3) (Kuze et al., 2009). $XCH_4$ is estimated by taking ratio of the total column amounts of $CH_4$ and the total column of dry-air which are extending from the Earth's surface to the top of the atmosphere.

The methodology to derive $XCH_4$ depends on the retrieval algorithm. For the NIES retrieval, profiles of the dry-air partial columns of $CO_2$, $CH_4$, $O_2$, and water vapor ($H_2O$) were simultaneously retrieved based on the maximum a posteriori (MAP) retrieval (Rodgers, 2000; Yoshida et al., 2013). The total column of dry-air is primarily derived from the surface pressure in consideration of the retrieved $H_2O$ profile and meteorological profiles from JMA (Yoshida et al., 2011, 2013). In case of the RemoTeC full-physics retrieval, operated at the Netherlands Institute for Space Research (SRON), The European Space

Agency (ESA), and at Heidelberg University, Germany, the dry-air column is calculated from ECMWF meteorological data (Butz et al., 2011). Another full-physics retrieval of the University of Leicester is based on the original Orbiting Carbon Observatory (OCO) retrieval and was modified for use with GOSAT spectra (UoL-OCFP) (Boesch and Noia, 2023). Furthermore, NIES, RemoTeC, and UoL-OCFP differ in the number of vertical layers and the aerosol parametrization which includes the number of aerosol types (Yoshida et al., 2013; Butz et al., 2011; Guerlet et al., 2013; Takagi et al., 2014; Boesch

and Noia, 2023).

In this study, we selected level 2 $XCH_4$ data in sun-glint mode from the NIES v02.95, the RemoTeC v2.3.8 full-physics retrieval from SRON, and the UoL-OCFP v7.3 (Copernicus Climate Change Service, Climate Data Store, 2018). A comparison with the RemoTeC v2.4.0 full-physics retrieval operated at Heidelberg University is shown in **Appendix A (Fig. A3)**. All data were bias corrected and cloud screened using the cloud flags obtained from the TANSO-Cloud and Aerosol

Imager (CAI) onboard GOSAT (Yoshida et al., 2011, 2013; Butz et al., 2011). In the following we refer to data obtained by the retrieval algorithm from NIES v02.95, RemoTeC v2.3.8, and UoL-OCFP v7.3 simply as "NIES", "RemoTeC", and "OCFP", respectively. The comparison with $XCH_4$ data retrieved from other satellites like GOSAT 2, launched in 2018, and TROPOMI, launched end of 2017, was not possible in our study due to missing aircraft data after mid-2017 **(section 2.1)**.

## 3 Methodology

### 3.1 Study region

**Figure 1** shows the study region and location of $CH_4$ in situ data. All data obtained over land are excluded. We selected the latitude–longitude ranges g1 = 30–40° N, 130–150° E and g2 = 20–30° N, 130–150° E of the western Pacific for the years from 2014 to end of 2017 for 2 reasons. First, we want to use the same years and region where we successfully derived ship-aircraft based column-average dry-air mole fraction of $CO_2$, previously (Müller et al., 2021). 10° × 20° latitude–longitude

boxes were chosen to obtain enough co-located data for the seasonal and interannual comparison with satellite retrievals, where g1 is expected to be stronger influenced by the emission outflow from land as compared to g2 **(Fig. 1)**. Second, the temporal and spatial coincident ship and aircraft $CH_4$ data are currently limited to the northern West Pacific until mid-2017 **(section 2.1)**. Within the two latitude–longitude boxes, we calculate monthly averages of the satellite and in situ



observations, and model results. The average number of monthly satellite observations at g1 and g2 of NIES is 28 ± 13 (24

months) and 34 ± 24 (31 months), of RemoTeC 24 ± 16 (11 months) and 41 ± 24 (24 month), and of OCFP 8 ± 2 (6 months)

and 14 ± 11 (27 months), respectively. Months with less than 5 observations are excluded. Ship observations of TF5 and

RYF from south and east of Japan were combined. On average, we obtained 6 ± 4 days of ship observations each month. The

number of monthly aircraft observations was 2 ± 1 for both latitude ranges. In our study, we develop the methodology for the

future application with higher numbers of in situ data.

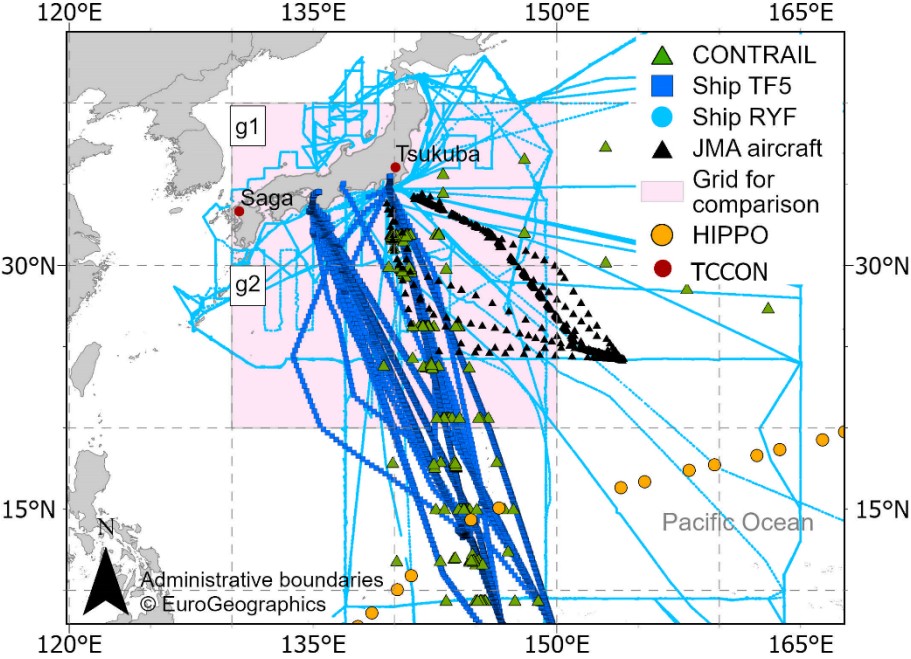

**Figure 1:** Location of $CH_4$ in situ data from aircraft (CONTRAIL: green triangles, JMA aircraft: black triangles), ship (Ship TF5: blue squares, Ship RYF: light blue circles) between 2014 and 2018. Also shown are the location of TCCON stations (red circles) and HIPPO profile flights (yellow circles). Selected regions within 10° × 20° latitude–longitude boxes are shown as pink shaded areas. Administrative boundaries © EuroGeographics.

### 3.2 Observation-based $CH_4$ profile construction and $XCH_4$ calculation

**Figure 2** illustrates the principle of how to construct ship–aircraft based $CH_4$ profiles from which $XCH_4$ is derived. Ship data are extrapolated vertically up to ~850 hPa, which represents the pressure level of the boundary layer above sea level. During summer, $CH_4$ is mainly removed by OH oxidation and has an instantaneous lifetime as short as 1 year (Patra et al., 2009, Saunois et al., 2020). In the same period, the $CH_4$ concentration can be increased in the mid-to-upper troposphere at the

western Pacific by $CH_4$ rich airmasses transported from South and East Asia (Umezawa et al., 2012). To constrain the tropospheric $CH_4$ variability, CONTRAIL aircraft data from the cruise portion of the flight at around 200 hPa, and JMA aircraft data from about 450 hPa are selected, which represents the upper and middle troposphere, respectively.



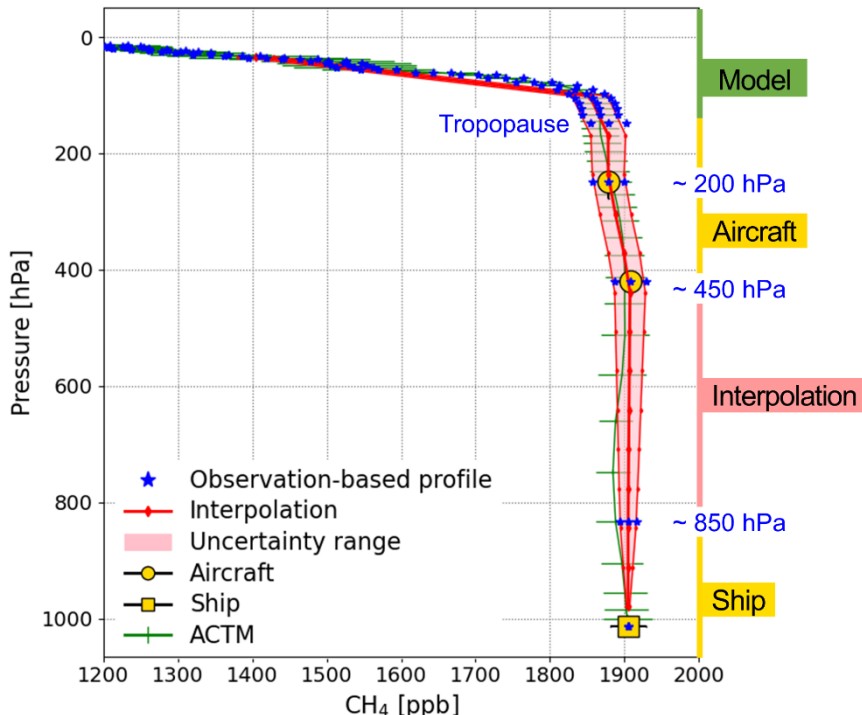

**Figure 2:** Construction of the observation-based CH₄ profile (blue) obtained by using ship and aircraft data (yellow) together with model
results (green), and the interpolation onto the pressure grid of the satellite retrieval (red). The example is obtained at the latitude 30–40° N,
in September 2015.

In the following, we test 3 approaches. Approach 1 is the adaptation of the approach of Müller et al. (2021). We extrapolate
CONTRAIL data upwards to the TROPPB and downwards to the lower cruising height at 400 hPa without the constraint of
the JMA aircraft data. Then we linearly interpolate in both pressure and dry-air mole fraction between the extrapolated ship
data, and the extrapolated aircraft data. Approach 2 is the addition of JMA aircraft data to the mid troposphere. We linearly
interpolate between the extrapolated ship data, and both aircraft data **(Fig. 2)**. In approach 3, we fill in model results between
the aircraft data of JMA and CONTRAIL of approach 2. Since CONTRAIL flies very close to the TROPPB, we do not fill
model data between CONTRAIL and the TROPPB. Above the TROPPB, we use model results in all three approaches. To
calculate the XCH₄ that the satellite would have seen given our constructed CH₄ profile, we first interpolate these profiles
onto the corresponding monthly averaged pressure grid of the satellite retrievals, then we use Eq. (15) of Connor et al.
(2008):

$$X_{CH4}^m = X_{CH4}^a + \sum_j h_j\, a_{CH4,j}\, (x_m - x_a)_j \,, \tag{1}$$

where $X_{CH4}^m$ is that XCH₄ which the satellite would report if it observed the constructed CH₄ profile $x_m$ (as a true profile).
Extracted from the satellite retrievals, $X_{CH4}^a$ is the a priori XCH₄, $h_j$ the pressure weighting function, $a_{CH4,j}$ the column
averaging kernel, and $x_a$ the a priori CH₄ profile. In this study, we use only the pressure grid and parameters of NIES. In the





following, we refer to the calculated $X_{CH4}^m$ as "simple observation-based XCH$_4$" (simple obs. XCH$_4$), "observation-based XCH$_4$" (obs. XCH$_4$), and "model blended observation-based XCH$_4$" (blended obs. XCH$_4$) for results of approach 1, 2 and 3, respectively.

### 3.3 Uncertainty assessment of obs. CH$_4$ profiles

Uncertainties in the obs. XCH$_4$ (simple, blended) are caused by the CH$_4$ profile construction: a) the inter- and extrapolation of the in situ data in the troposphere, b) the tropopause height, and c) the modelled stratospheric column.

### 3.3.1 Tropospheric uncertainty

First, to assess the uncertainty due to the inter- and extrapolation, we investigated the variability of the CH$_4$ dry-air mole fractions observed by profile flights of the HIPPO number 4 campaign (HIPPO 4) over the Pacific Ocean (Wofsy, 2011). 240 Between 14 June to 11 July 2011, 20 profiles ranging from the surface up to about 13 km were obtained near the study region (**Fig. 1**). Within each profile, the CH$_4$ dry-air mole fractions show variations between 9–62 ppb (24 ± 17 ppb, **Table 1**). The highest range was seen in the middle to upper troposphere during these summer months, consistent with observations by Umezawa et al. (2012). Based on this variation, we use 24 ppb uncertainty between the extrapolated ship data and the TROPPB for the profile construction in approach 1.

**Table 1:** Uncertainty assessment of the obs. CH$_4$ profiles at the troposphere. Top rows: average concentration range of CH$_4$ within each HIPPO 4 profile (mean variability ± standard deviation). Bottom rows: average difference between MIROC4-ACTM (ACTM) and HIPPO 4 data (mean difference ± standard deviation of differences) at different altitude ranges.

| HIPPO 4 profile range [m] | Variation within profiles [ppb] |
|---|---|
| ~300–~13000 | 24 ± 17 |

| Altitude [m] | ACTM − HIPPO4 [ppb] |
|---|---|
| 0–1500 | 12 ± 3 |
| 1500–6000 | 0 ± 11 |
| 6000–11000 | 6 ± 15 |

Second, we assessed how good the MIROC4-ACTM reproduces the variation of 6 selected HIPPO profiles near the study 250 area (**Fig. 1**). We choose the MIROC4-ACTM to be consistent with our previous study (Müller et al., 2021). We distinguished the altitude range 0–1500 m, corresponding to the boundary layer, 1500–6000 m, corresponding to the middle troposphere between the extrapolated ship and JMA aircraft data, and 6000–11000 m, corresponding to the upper troposphere between the JMA and CONTRAIL aircraft data. The average difference between the MIROC4-ACTM and the HIPPO profiles were 12 ± 3 ppb, 0 ± 11 ppb, and 6 ± 15 ppb for the altitude ranges 0–1500 m, 1500–6000 m, and 6000–



11000 m, respectively (**Table 1**). Considering the maximum possible difference, we added 11 ppb uncertainty between the extrapolated ship and JMA data, and 21 ppb between the JMA data and up to the TROPPB in approach 2 and 3.

### 3.3.2 Tropopause uncertainty

The variation of the monthly averaged TROPPB (**section 2.3**) at 30–40° N was more than twice that at 20–30° N with an average standard deviation of $68 \pm 22$ hPa and $23 \pm 9$ hPa, respectively (**Table 2**). The maximum difference of 90 hPa (68

+ 22 hPa) and 32 hPa (23 + 9 hPa) at the level of the TROPPB corresponds to an altitude difference of 3 to 4 km, and 1 to 2 km, respectively. To test the impact of the TROPPB on the derived $XCH_4$, we first calculated the simple obs. $XCH_4$. Second, we calculated the simple obs. $XCH_4$ with TROPPB $\pm$ 90 hPa at 30–40° N and TROPPB $\pm$ 32 hPa at 20-30° N, based on the monthly averaged variability of the TROPPB. Then, we compared the latter two results with the original simple obs. $XCH_4$. The average difference in the resulting $XCH_4$ at 30–40° N and 20–30° N for the reduced TROPPB (−90 hPa, −32 hPa) was

$-4 \pm 3$ ppb and $-1 \pm 1$ ppb, respectively. If the TROPPB was increased (+90 hPa, +32 hPa), the difference was small as $1 \pm 2$ ppb and $0.1 \pm 0.2$ ppb (**Table 2**). Because model results are used above the TROPPB, a "too high" TROPPB (= too low altitude), can be compensated by the model. In total, the TROPPB causes an uncertainty of less than 0.4% on the calculated $XCH_4$.

**Table 2:** Uncertainty assessment of the obs. $CH_4$ profile. **a)** at the blended tropopause pressure (TROPPB) by calculating the difference
"simple obs. $XCH_4$ − simple obs. $XCH_4$ + reduced/ increased TROPPB ($XCH_{4\ (\pm TROPPB\ var)}$)" (mean difference $\pm$ standard deviation of differences). TROPPB var = monthly average variability of TROPPB (mean standard deviation of the monthly averages $\pm$ standard deviation). **b)** at the stratospheric column by calculating the difference "simple obs. $XCH_4$ − simple obs. $XCH_4$ with extrapolated aircraft data up to 0.0128 hPa ($XCH_{4(no\_str)}$)", and the differences between the modelled stratosphere of MIROC4-ACTM (ACTM), CAMS, and CAMSinv (mean difference $\pm$ standard deviation of differences).

| a) Tropopause pressure (TROPPB) | Latitude 30–40°N | Latitude 20–30°N |
|---|:---:|:---:|
| monthly average TROPPB variation (TROPPB var) [hPa] | $68 \pm 22$ | $23 \pm 9$ |
| simple obs. $XCH_4 - XCH_{4\ (-TROPPB\ var)}$ [ppb] | $-4 \pm 3$ | $-1 \pm 1$ |
| simple obs. $XCH_4 - XCH_{4\ (+TROPPB\ var)}$ [ppb] | $1 \pm 2$ | $0.1 \pm 0.2$ |
| **b) Stratosphere** | | |
| simple obs. $XCH_4 - XCH_{4(no\_str)}$ [ppb] | $-37 \pm 5$ | $-26 \pm 5$ |
| ACTM − CAMS [ppb] | $-138 \pm 9$ | $-165 \pm 15$ |
| ACTM − CAMSinv [ppb] | $23 \pm 5$ | $24 \pm 7$ |




### 3.3.3 Stratospheric uncertainty

$CH_4$ shows variations in the stratosphere due to its reactions with excited oxygen ($O(^1D)$), OH and chlorine radicals (Saunois et al., 2020), which is represented in each model differently. GOSAT NIES $CH_4$ observations have a higher sensitivity in the stratospheric column as compared to $CO_2$ (averaging kernel >0.8 in the stratosphere, **Appendix A, Fig. A1**). Therefore, the

shape and value of the modelled stratospheric $CH_4$ column impact the derived column-averaged dry-air mole fractions more than those for $CO_2$.

In the first step, we used the simple obs. $XCH_4$ to test the sensitivity of the stratospheric column on the derived $XCH_4$. We extrapolated CONTRAIL aircraft data through the TROPPB and the stratosphere up to 0.0128 hPa. $XCH_4$ calculated from profiles without considering the stratosphere was higher than the simple obs. $XCH_4$ by $37 \pm 5$ ppb ($2.0 \pm 0.3$ %) and

$26 \pm 5$ ppb ($1.4 \pm 0.3$ %) at 30–40° N and 20–30° N, respectively (**Table 2**), which confirms the importance of the stratospheric column to derive $XCH_4$ correctly.

In the second step, we assessed the uncertainty of the stratospheric model. We calculated the difference MIROC4-ACTM − CAMS and MIROC4-ACTM − CAMSinv of the monthly averaged data above the TROPPB. Then, we interpolated the MIROC4-ACTM data with its higher resolved pressure grid on that of the CAMS and CAMSinv data, respectively (**section**

**2.3**). CAMS was positively biased by $138 \pm 9$ ppb, and $165 \pm 15$ ppb at 30–40° N and 20–30° N, respectively (**Table 2, Appendix A, Fig. A2 (a), (b)**). In contrast, the total average difference between MIROC4-ACTM and CAMSinv was small as $23 \pm 5$ ppb, and $24 \pm 7$ ppb at 30–40° N and 20–30° N, respectively. In addition, the difference MIROC4-ACTM − CAMSinv depends on the season. The highest average difference occurred in June (30–40° N: $37 \pm 6$ ppb, 20–30° N: $44 \pm 3$ ppb), the lowest in October ($4 \pm 0.6$ ppb) at 30–40° N, and January ($5 \pm 5$ ppb) and February ($3 \pm 13$ ppb) at 20–30° N

(**Appendix A, Fig. A2 (c), (d)**). A large positive stratospheric $CH_4$ bias of around 200 ppb of CAMS was recently reported by Agustí-Panareda et al. (2023), consistent with our observations. They suggest that uncertainties associated with the stratospheric chemical loss of $CH_4$ are the largest contributor to that bias. Compared to CAMS, both the MIROC4-ACTM and CAMSinv use optimized atmospheric transport models and account for chemical losses in the stratosphere. The seasonality of the difference MIROC4-ACTM − CAMSinv indicates that the seasonal dependent chemical loss of $CH_4$

and/or meridional transport processes are modelled differently in both models.

Based on the total average difference between the latter 2 models, we added a $\pm 24$ ppb uncertainty to the stratospheric column of the constructed $CH_4$ profile. The impact of the three stratospheric models on the calculated $XCH_4$ using this uncertainty is discussed in **section 4.2**.





# 4 Results and Discussion

## 4.1 Evaluation of the approaches

**Figure 3** shows the temporal variation of the XCH$_4$ calculated for the two selected latitude ranges (g1 = 30–40° N, g2 = 20–30° N) using approach 1 (simple obs. XCH$_4$), approach 2 (obs. XCH$_4$), and approach 3 (blended obs. XCH$_4$). For the period 2014 to mid-2017, we obtained 20 and 31 monthly averaged XCH$_4$ at the latitude ranges 30–40° N and 20–30° N, respectively. The uncertainty range of the simple obs. XCH$_4$ (22 ppb) is by 5 ppb larger than those of the obs. XCH$_4$ and blended obs. XCH$_4$ (17 ppb) (**section 3.3**). Furthermore, the difference between the latter 2 approaches is as small as 1 ± 3 ppb (blended obs. XCH$_4$ − obs. XCH$_4$) at both latitude ranges. In contrast, the difference between simple and blended obs. XCH$_4$ shows a variability of 2 ± 11 ppb and 4 ± 9 ppb at 30–40° N and 20–30° N, respectively.

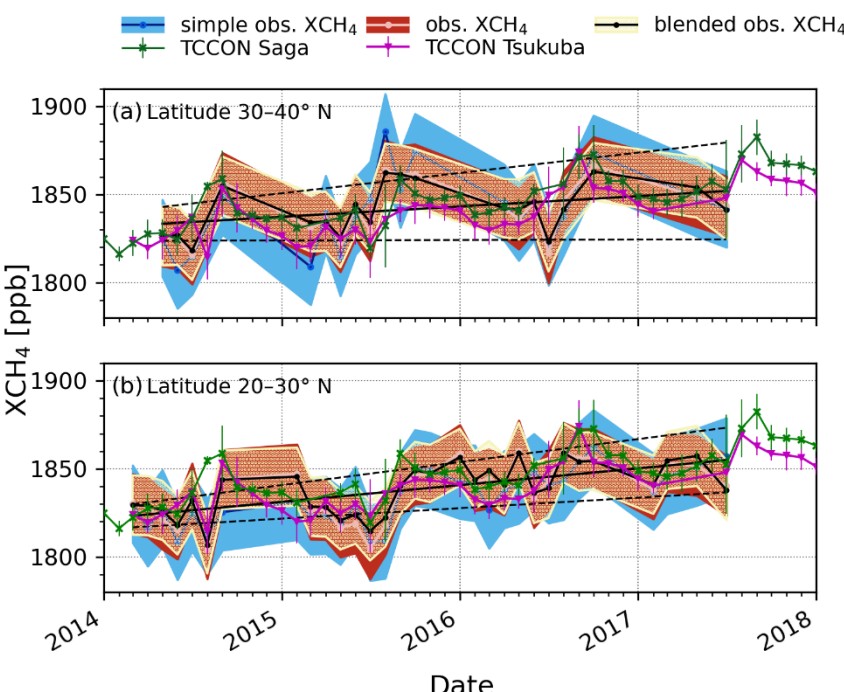

**Figure 3:** Temporal variation of monthly averaged XCH$_4$ obtained by approach 1 (simple obs. XCH$_4$, red), approach 2 (obs. XCH$_4$, blue), and approach 3 (blended obs. XCH$_4$, black), and from the TCCON station in Saga (green) and Tsukuba (magenta) at the latitude range 30–40° N (a) and 20–30° N (b). Shaded areas are the uncertainty ranges of the approaches; error bars are the standard deviations of TCCON. Also shown is the linear least-square regression (thick black line) with a 90% confidence interval on the slope and intercept (black dashed line) of approach 3.

To assess the correctness of the XCH$_4$ datasets, we compare our data at both latitude ranges with the monthly averaged XCH$_4$ data (version GGG2020) obtained from the nearest ground based TCCON stations in Tsukuba (36.05° N, 140.12° E, Morino et al., 2022) and Saga (33.24° N, 130.29° E, Shiomi et al., 2022) **(Fig. 1, 3)**. Compared to Tsukuba, Saga is influenced by the continental outflow of airmasses from East Asia. It is noted that the distance of about 1300 km between the





TCCON stations and the center of g2 is large. Considering that there are no strong $CH_4$ sources over the open ocean at g2, the comparison gives us an indication about the applicability of the datasets.

By looking at the averaged difference at g1, $XCH_4$ from Tsukuba was lower than that derived from our approaches with $-3 \pm 20$ ppb, $-3 \pm 14$ ppb, and $-4 \pm 13$ ppb for approach 1, 2, and 3, respectively. In contrast, $XCH_4$ from Saga was higher and showed better agreement with differences of $1 \pm 20$ ppb for approach 1, and $1 \pm 12$ ppb for approach 2 and 3. At g2, $XCH_4$ from Tsukuba matches our data better than that from Saga with $3 \pm 11$ ppb, $0 \pm 12$ ppb, and $-1 \pm 11$ ppb for approach 1, 2, and 3. Saga showed a higher discrepancy of $9 \pm 12$ ppb, $6 \pm 14$ ppb, and $5 \pm 14$ ppb for the respective approaches. The

similarity between $XCH_4$ from our approaches and Saga at g1, and Tsukuba at g2, indicates that the ocean area at 30–40° N (g1) is rather influenced by the continental outflow of $CH_4$ from Asia, while 20–30° N (g2) showed cleaner conditions.

Given the lower uncertainty range and lower maximal possible averaged difference between TCCON and approach 2 and 3 compared to approach 1, the latter 2 approaches are preferable for future applications. A clear advantage of the more elaborate approach 3 is not seen but might become clearer with extended in situ datasets. Therefore, we use the results of

approach 3 (blended obs. $XCH_4$) for further discussion.

**4.2 Evaluation of the stratospheric model**

**Figure 4** shows the comparison of the blended obs. $XCH_4$ (approach 3) using the MIROC4-ACTM, CAMS, and CAMSinv for the stratospheric column (**section 3.3.3**), denoted as $ACTM_{XCH4}$, $CAMS_{XCH4}$, and $CAMSinv_{XCH4}$. Using $ACTM_{XCH4}$ as reference, $CAMS_{XCH4}$ is highly biased at both latitude ranges by $12 \pm 5$ ppb ($0.6 \pm 0.2\%$) in total. In contrast, $CAMSinv_{XCH4}$

shows a small negative total bias of $-5 \pm 3$ ppb ($-0.3 \pm 0.2\%$). The similarity of the latter two results and their differences to $CAMS_{XCH4}$ shows the strong impact of the stratospheric part on the derived $XCH_4$ and highlights the importance to make an appropriate model choice (**compare section 3.3.3**). Based on our observations, we suggest using either the MIROC4-ACTM or CAMSinv to model the stratospheric column. In the following, we use the $ACTM_{XCH4}$ to demonstrate the applicability of the dataset for satellite evaluation. However, for the operational application in future, the public available CAMSinv might

be the better choice until the MIROC4-ACTM will be available in near-real time.

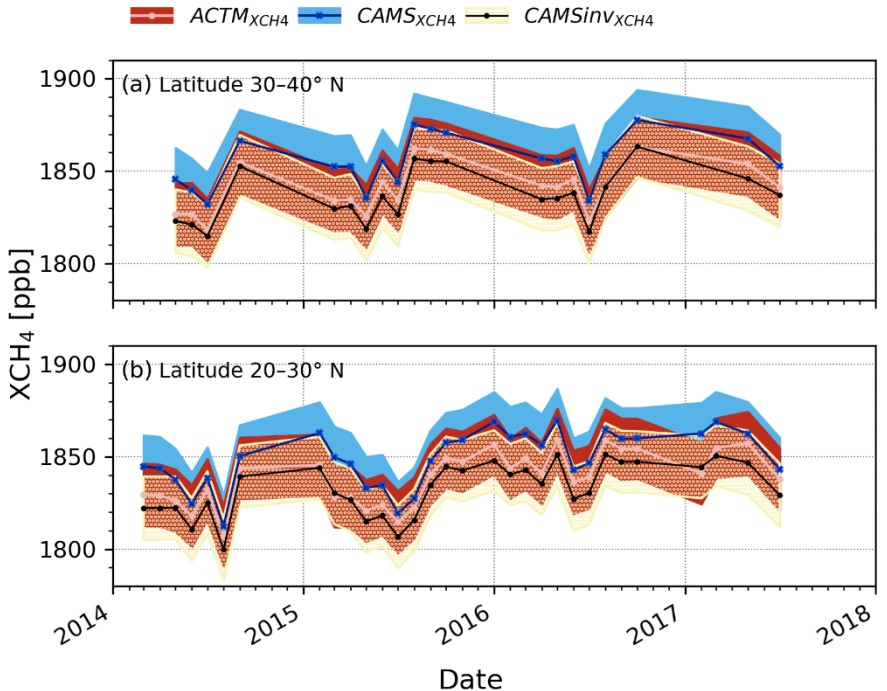

**Figure 4:** Comparison between the blended obs. XCH$_4$ (approach 3) derived from CH$_4$ profiles using the MIROC4-ACTM (ACTM$_{XCH4}$, red), CAMS (CAMS$_{XCH4}$, blue), and CAMSinv (CAMSinv$_{XCH4}$, black) for the stratospheric column at the latitude range 30–40° N (a) and 20–30° N (b). Shaded areas are the uncertainty ranges.

## 4.3 Applicability of observation-based XCH$_4$

In the following we want to demonstrate the applicability of the in situ derived XCH$_4$ datasets for carbon cycles studies by analyzing the seasonal variation of XCH$_4$ over the ocean, and for satellite evaluation. We will focus on the blended obs. XCH$_4$.

### 4.3.1 Seasonal variation

**Figure 3** shows that all three approaches follow similar temporal variations and trends. At 30–40° N, a rough seasonal cycle with lower values between winter and summer (minima in July) and maxima between August to October is seen. The column observations of CH$_4$ are consistent with northern hemispherical surface observations (e.g., Dlugokencky et al., 1995). Minima between July and August and maxima in the period winter to spring have been observed at the lower troposphere by aircraft and ground based stations in Japan (Umezawa et al., 2014; Tohjima et al., 2002). The seasonal characteristics are explained by the interaction between air mass origin and atmospheric OH concentration. In summer, south-easterly air masses from CH$_4$ source-free regions of the Pacific Ocean and the surrounding of Japan are dominant. In addition, the OH concentration is highest in summer, which leads to enhanced CH$_4$ removal from the atmosphere. In winter,



when the removal through OH oxidation is lowest, prevailing north-westerly winds bring $CH_4$ rich air masses from China and Siberia (Umezawa et al., 2014; Tohjima et al., 2002). This explains the larger $CH_4$ concentration during that period.

At 20–30° N, lower values are obvious from winter to the end of summer (August) in 2015, but in 2016, it is not as clear as at 30–40° N. **Figure 3** also shows the linear least-squares regression with 90% confidence interval on the slope and intercept. At 30–40° N, the annual increase in $XCH_4$ is within the uncertainty range with $9 \pm 9$ ppb for the simple obs. $XCH_4$, and $6 \pm 6$ ppb for the other two approaches. In contrast, at 20–30° N, the annual increase is significant with $11 \pm 3$ ppb for the simple obs. $XCH_4$, and $10 \pm 4$ and $9 \pm 4$ ppb for the obs. and blended obs. $XCH_4$. The higher summertime values in 2016 contribute

to the difference in the growth rates at 20–30° N. Similar strong growth rates have been reported for the global atmospheric $CH_4$ concentration between 2014 to 2017 with a peak in 2014 of 13 ppb, and a minimum in 2016 of 7 ppb (Nisbet et al., 2019). It is noted that limited and uneven sampled in situ data during each month might cause an artificial difference between the latitude ranges.

A possible explanation for the observed increased summertime $XCH_4$ values in 2016 can be the characteristics of the

prevailing southerly winds in that season. In the years 2015 to 2016, a strong El Niño event took place, which is linked to extreme heat and drought, and consequently to increased biomass burning in tropical regions (Bousquet et al., 2006; Parker et al., 2016; Whitburn et al., 2016). Smoldering combustion in peatland fires can release large amounts of $CH_4$ (Bousquet et al., 2006; Parker et al., 2016). Using GOSAT observations, Parker et al. (2016) reported an enhancement of $XCH_4$ of 35 ppb above background conditions over Indonesian peatland fires at the end of 2015. Furthermore, Zhang et al. (2018)

demonstrated that enhanced $CH_4$ emissions from wetland areas from January through May 2016 are related to the strong El Niño which provides an explanation for a rise of the atmospheric $CH_4$ growth rate. Therefore, southerly air masses of high $CH_4$ concentration might have affected the study region. Our observations demonstrate the capability of the ship-aircraft based dataset to capture seasonal variations and climatological events like the El Niño.

**4.3.2 Satellite evaluation**

**Figure 5** shows the temporal variation of the blended obs. $XCH_4$ ($ACTM_{XCH4}$) in comparison with $XCH_4$ from GOSAT observations using the NIES, RemoTeC, and OCFP retrieval (**section 2.4**). The retrievals mostly lie in the uncertainty range (17 ppb) of the blended obs. $XCH_4$. The difference "blended obs. $XCH_4$ − NIES" is $-0.04 \pm 12.65$ ppb and $-0.04 \pm 13.32$ ppb at 30–40° N and 20–30° N, respectively. The high standard deviations are similar to that reported for the difference between NIES and TCCON ocean data in the data release note of the NIES GOSAT project (NIES GOSAT Project, 2020).

The difference between blended obs. $XCH_4$ and RemoTeC show a larger average discrepancy of $11.8 \pm 16.2$ ppb and $12.1 \pm 8.1$ ppb but with smaller standard deviation at 20–30° N. At 30–40° N, OCFP provides almost no valid data. The difference is $2.2 \pm 21.0$ ppb. At 20–30° N, the difference "blended obs. $XCH_4$ – OCFP" of $10.3 \pm 9.6$ ppb is similar to the difference of RemoTeC. The smaller standard deviations of RemoTeC and OCFP suggest rather a systematic offset at that latitude range. The higher difference compared to NIES can arise from the choice of a priori profiles and column averaging kernel in the



retrieval and their choice in calculation of the blended obs. XCH₄ (**section 3.2**). To clarify if the offset of RemoTeC and

OCFP is a true regional or ocean bias, further analyses are needed in future.

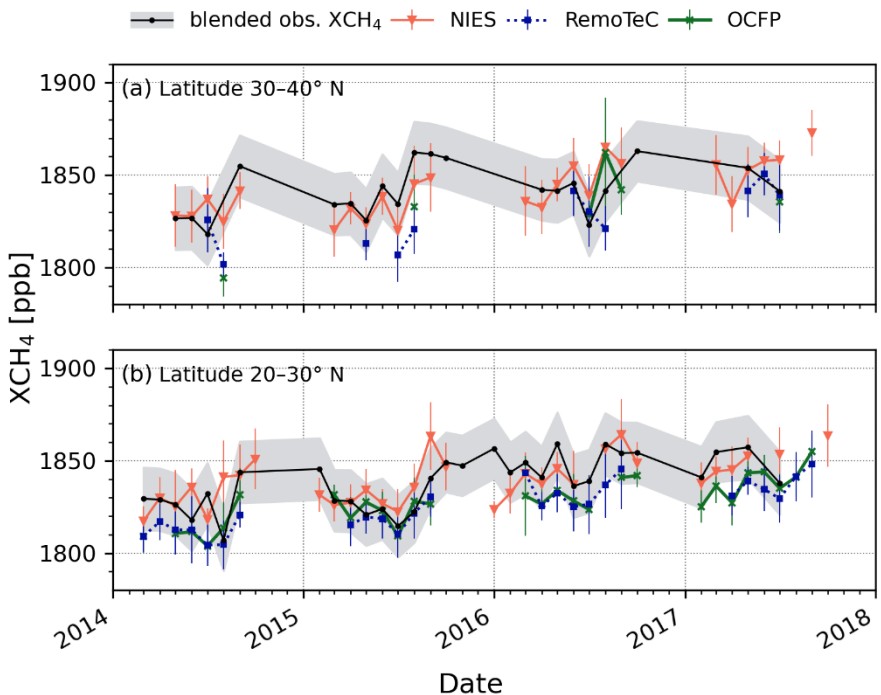

**Figure 5:** Temporal variation of the blended obs. XCH₄ (ACTM$_{XCH4}$, black) in comparison with GOSAT XCH₄ retrievals from NIES (red), RemoTeC (blue), and OCFP (green) at the latitude range 30–40° N (a) and 20–30° N (b).

**Conclusion**

As reference dataset for satellite validation and carbon cycle studies, we investigated three different approaches to derive column-averaged dry-air mole fractions of $CH_4$ ($XCH_4$) over oceans by integrating commercial ship and aircraft observations. The study focused on the latitude ranges 30–40° N and 20–30° N at the longitude 130–150° E between the years 2014 and 2018. Approach 1 used simple linear inter- and extrapolation between ship and aircraft data of the upper

troposphere; approach 2 used additional aircraft data of the middle troposphere, and approach 3 added model results between the middle and upper tropospheric aircraft observations. All three approaches used model results for the stratospheric column.

Uncertainties of the calculated $XCH_4$ were reduced by 5 ppb from 22 ppb (approach 1) to 17 ppb for approach 2 and 3. $XCH_4$ derived from approach 2 and 3 were similar within 1 ± 3 ppb. The difference between approach 3 and 1 was about

30% higher. At 30–40° N, $XCH_4$ data of the TCCON station Saga, influenced by the Asian continental outflow, showed a better agreement with our approaches (within 1 ± 20 ppb for approach 1, 1 ± 12 ppb for approach 2 and 3) than that from Tsukuba (which was lower by −3 ± 20 ppb, −3 ± 14 ppb, and −4 ± 13 ppb than approach 1, 2, and 3). At 20–30° N, better


agreement was found with TCCON data of Tsukuba (difference of Tsukuba: $3 \pm 11$ ppb, $0 \pm 12$ ppb, and $-1 \pm 11$ ppb, and of Saga of: $9 \pm 12$ ppb, $6 \pm 14$ ppb, and $5 \pm 14$ ppb, for approach 1, 2, and 3). These observations indicate a stronger impact of

continental emissions on the higher latitudinal study area. Based on the uncertainty and difference towards TCCON, we selected approach 3, defined as blended observation-based $XCH_4$ (blended obs. $XCH_4$).

Applying approach 3, we found that omitting the stratospheric column in the $CH_4$ profile impacts the derived blended obs. $XCH_4$ by about 2%, which is significantly higher than the corresponding impact on the derived $XCO_2$ of our previous study (<0.1%). Using CAMSinv or MIROC4-ACTM for the stratospheric column, the derived blended obs. $XCH_4$ was similar

within 8 ppb ($0.3 \pm 0.2$%). Using CAMS instead of MIROC4-ACTM, the blended obs. $XCH_4$ was higher biased by $12 \pm 5$ ppb ($0.6 \pm 0.2$%). Since MIROC4-ACTM and CAMSinv use optimized transport models and consider chemical losses in the stratosphere, we propose either to use CAMSinv or MIROC4-ACTM, of which CAMSinv is publicly available.

The temporal variation of the blended obs. $XCH_4$ showed minima in summer (July) and maxima between August and October, and an annual growth rate between 6 and 10 ppb, consistent with previous studies. In 2016, we observed a weaker

summertime minimum and suggest that this is the result of the strong 2015/2016 El Niño event which was related to higher $CH_4$ emissions and growth rates. The comparison of our results with GOSAT $XCH_4$ retrievals from NIES showed strong scatter of the differences with $-0.04 \pm 13$ ppb. In contrast, RemoTeC and OCFP showed a larger but rather systematic negative bias of $-12.1 \pm 8.1$ ppb and $-10.3 \pm 9.6$ ppb at 20–30° N, which is likely related to differences in a priori profiles and column averaging kernels of the retrieval.

We are aware that our results are based on very limited in situ data, and conclusions made from the seasonal variation and the quantitative descriptions need to be taken carefully. We want to stress that in the near future aircraft observations by CONTRAIL over the western Pacific Ocean will re-start, probably within the next 2 years. Together with data of other aircraft projects like that of the In-service Aircraft for a Global Observing System (IAGOS) project, $CH_4$ data of the upper troposphere will increase rapidly. However, additional ship routes would be beneficial. Despite the current limitations, our

study could demonstrate the applicability of the new dataset as reference for carbon cycle studies and for satellite evaluation over oceans. As a complement to established validation networks, we can contribute with our ship-aircraft derived $XCH_4$ dataset to the validation of TROPOMI, GOSAT-GW and other upcoming satellite missions in future.





**Appendix A**

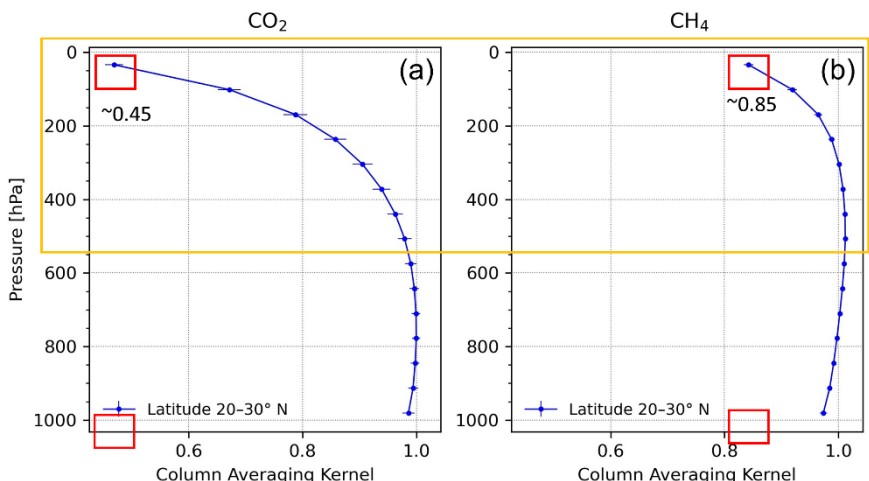

**Figure A1:** The GOSAT NIES column averaging kernel (ak) in dependence of the pressure for $CO_2$ (a) and $CH_4$ (b) at the latitude range 20–30° N. The yellow square indicates the area of major differences; the red squares emphasize the difference in the ak value at the lowest pressure of 34 hPa. Compared to the ak of $CO_2$, the impact of the $CH_4$ profile on the calculated $XCH_4$ is high below the tropopause (400 – 200 hPa) and at the stratospheric part.

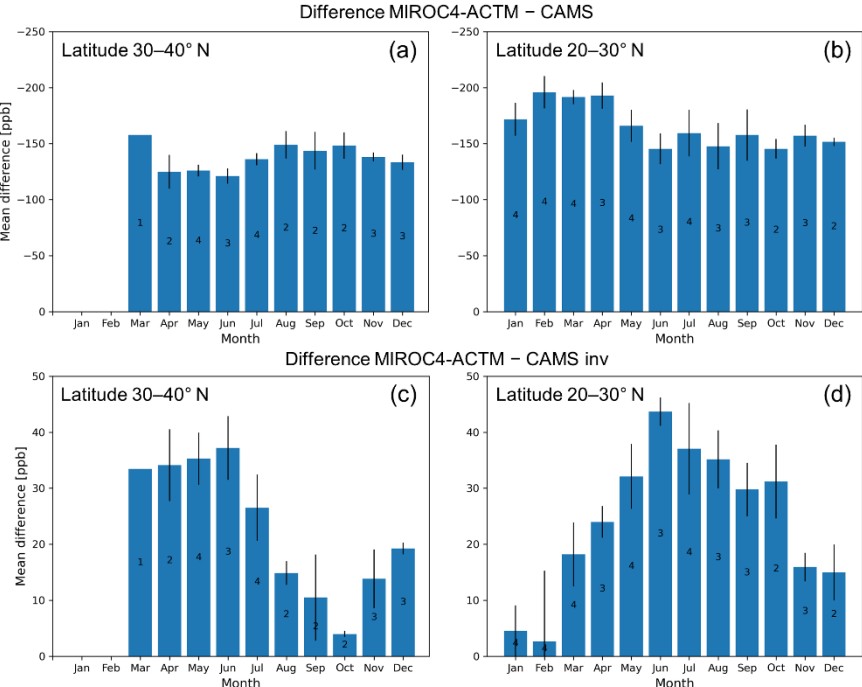

**Figure A2:** Monthly averaged difference between MIROC4-ACTM and CAMS (a), (b), and MIROC4-ACTM and CAMSinv (c), (d) at the latitude range 30–40° N and 20–30° N, respectively. Error bars are the standard deviation of the monthly averages. Numbers inside the bars correspond to the number of mean values per month.



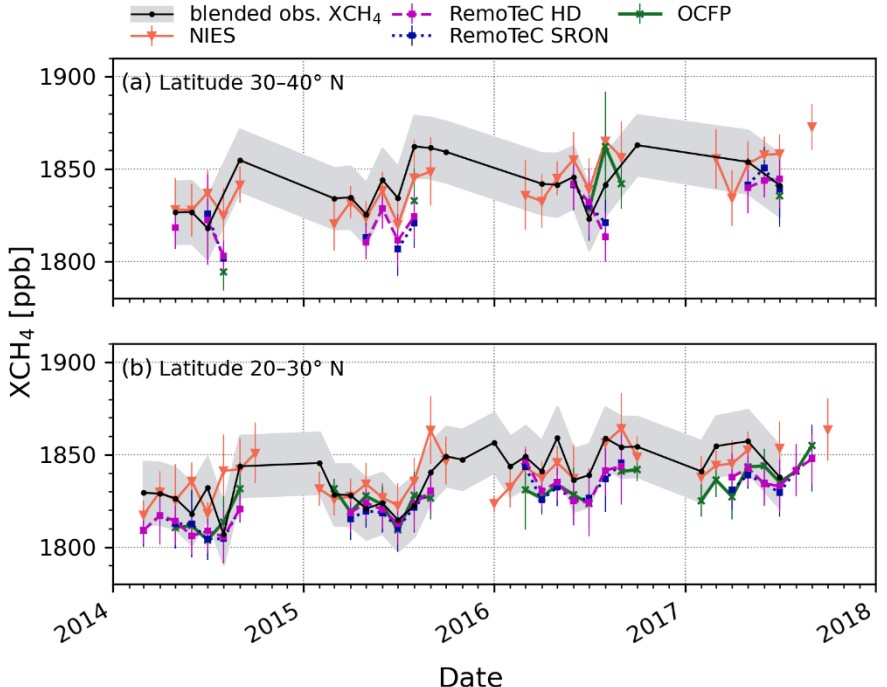

**Figure A3:** Temporal variation of the blended obs. XCH$_4$ (ACTM$_{XCH4}$, black) in comparison with GOSAT XCH$_4$ retrievals from NIES (red), RemoTeC Heidelberg (HD) (magenta), RemoTeC SRON (blue), and OCFP (green) at the latitude range g1 = 30–40° N (a) and g2 = 20–30° N (b). The difference RemoTeC HD − RemoTeC SRON is −0.4 ± 4.4 ppb and 1.6 ± 3.0 ppb (mean difference ± standard deviation of differences) at g1 and g2, respectively. The average number of valid retrievals per month for RemoTeC HD (g1: 20 ± 14 ppb, 13 months; g2: 50 ± 29 ppb, 24 months) is larger than for RemoTeC SRON (g1: 24 ± 16 ppb, 11 months; g2: 41 ± 24 ppb, 24 months).

**Data availability**

The GOSAT data of the NIES retrieval algorithm are available from the GOSAT Project website of the National Institute for Environmental Studies (NIES) at https://data2.gosat.nies.go.jp/index_en.html, last access: 17 May 2023.

GOSAT data of the RemoTeC full-physics retrieval from SRON (SRFP) and the OCO full-physics retrieval by the University of Leicester (OCFP) are available from the Copernicus Climate Change Service, Climate Data Store at https://doi.org/10.24381/cds.b25419f8, accessed on 17 May 2023.

XCH$_4$ data of the RemoTeC full-physics retrieval by Heidelberg University are available upon request (andre.butz@iup.uni-heidelberg.de).

The CH$_4$ mole fraction data of CONTRAIL (https://doi.org/10.17595/20190828.001, Machida et al., 2019) are available from the Global Environmental Database (GED) of NIES (https://db.cger.nies.go.jp/ged/en/links/index.html?id=link1, GED, 2022). CONTRAIL data are also available from the World Data Center for Green-house Gases (WDCGG) at https://gaw.kishou.go.jp/, last access: 17 May 2023. NIES SOOP CH$_4$ will be released at the GED by the end of 2023.



CAMS and CAMSinv data are available from the Atmosphere Data Store operated by the European Centre for Medium-Range Weather Forecasts at https://ads.atmosphere.copernicus.eu, last access: 17 May 2023.

TCCON data are available from the TCCON Data Archive hosted by CaltechDATA at https://tccondata.org, last access: 17 May 2023.

MIROC4-ACTM concentration data are available upon request (prabir@jamstec.go.jp).

**Supplement**

Data used in this study accompany the article.

**Author contribution**

The study was designed by HT. Data analyses were made by AM. MIROC4-ACTM data were provided by PKP. CAMSinv data were provided by CWO. Extensive discussions were made by AM, HT, TS, PKP. The paper was written, edited, and proofed by all the authors.

**Competing interests**

The co-author André Butz is an Executive editor of the editorial board of the AMT. The authors have no other competing interests to declare.

**Acknowledgements**

We are grateful to Christopher W. O'Dell from the Colorado State University who extracted and provided the CAMS global inversion-optimized greenhouse gas fluxes and concentrations dataset (CAMSinv).

The authors acknowledge the satellite data infrastructure for providing access to the GOSAT NIES data.

We acknowledge the Copernicus Climate Change Service (C3S) Climate Data Store (CDS) for providing access to the GOSAT SRFP and OCFP data. We thank Michael Buchwitz and Hartmut Bösch from the University of Bremen for the project management of the methane SRFP and OCFP data, Robert Parker from the University of Leicester for the data generation and Antonio Di Noia from University Bremen for the FP data.

We also acknowledge the Copernicus Atmosphere Monitoring Service (CAMS) operated by the European Centre for Medium-Range Weather Forecasts on behalf of the European Commission as part of the Copernicus program.



We thank Japan Meteorological Agency (JMA) and the World Data Centre for Greenhouse Gases (WDCGG) for providing aircraft data by Kazuyuki Saito (JMA) and ship data of the Ryofu Maru, R/V by Kazutaka Enyo (JMA) and Koji Kadono (JMA).

We acknowledge the TCCON science team at Tsukuba and Saga. The TCCON station at Tsukuba is supported in part by the GOSAT series project.

Observational projects of CONTRAIL and NIES SOOP are financially supported by the research fund of the Global Environmental Research Coordination System of the Ministry of the Environment, Japan.

This research is a contribution to the Research Announcement on GOSAT series joint research, titled "Combined cargo-ship

and passenger aircraft observations-based validation of GOSAT-2 GHG observations over the open oceans", the GOSAT-GW NIES mission, and to the Greenhouse Gas Initiative of the Atmospheric Composition Virtual Constellation (AC-VC) of the Committee on Earth Observation Satellites (CEOS). We also would like to thank the anonymous reviewers for the valuable comments and suggestions for improving the paper.

**Financial support**

This research has been supported mainly by the Global Environmental Research Coordination System from the Ministry of the Environment (Japan) (grant nos. E1851, E1253, E1652, E1151, E1951, E1451, E1751, E1432, E2151, and E2252), and partly by the Environmental Research and Technology Development Fund (ERTDF) of the Environmental Restoration and Conservation Agency provided by Ministry of the Environment of Japan (grant nos. 2-1803 and 2-2201, JPMEERF20182003 and JPMEERF20222001).

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
