# Peer review of "Ship- and aircraft-based XCH4 over oceans as new tool for satellite validation"

_Atmospheric Measurement Techniques, 2023_

## Author Comment (AC1)

**General.**

We would like to thank the anonymous Referee #1 for providing comments to improve and clarify our manuscript. We will revise the text by fully taking the comments into account. Please find our responses to the specific comments and questions below. Our response is written in bold. The revised parts of the manuscript are highlighted in bold red.

5 ## Comments of Referee #1 and our responses to them

**General comments**

This paper combines surface and aircraft measurements of atmospheric methane, together with modeling estimates to generate a reference dataset of methane column over a large region over the pacific ocean, south-east of Japan. The resulting time series are analyzed to discuss the growth rate and seasonal cycle.

10 The paper is very well written. The method is clearly described and the various uncertainties are discussed in detail. The paper can be published with minimal changes. I nevertheless offer some suggestions to the authors below :

**Specific comments**

Comment 1

Line 27: I recommend to use ppb, consistently with the rest of the text, rather than %

15 **Response**

**We added the value in ppb to be consistent with the rest of the text. The revised sentence is as follows:**

Lines 26–27: *Depending on the models, the difference can be more than **12 ppb** (0.**6** %), showing the importance for the appropriate choice.*

Comment 2

20 Line 36: You could say that methane is the second most important anthropogenic GHG after CO2 (rather than "one of the most")

**Response**

**Thank you, we clarified the sentence as follows:**

Lines 37–38: *Methane ($CH_4$) is the **second** most important anthropogenic greenhouse gas (GHG) in the atmosphere **after***
25 *carbon dioxide ($CO_2$).*

Line 208: "Instantaneous lifetime as short as one year" for the summer condition is not clear. Rather, you could provide the oxydatation fraction per month

 **Response**

30   **Thank for pointing this out. The expression "Instantaneous lifetime" is often used by modelers. Compared to the global atmospheric lifetime, it describes the lifetime at a specific time and location. The global atmospheric lifetime of $CH_4$ is 9.1 years (Szopa et al., 2021). But looking at the troposphere of the northern midlatitudes during the summer month July, the lifetime of $CH_4$ can be short as 1 year, as shown in Fig. 14 of Patra et al. (2009) below. During the same month, the instantaneous lifetime of $CH_4$ of the southern hemisphere is longer.**

35   **Figure 14 of Patra et al. (2009) illustrates the different instantaneous lifetimes in boreal winter, upper plot (a), in comparison with boreal summer, lower plot (b). At 30° N, the lifetime at the lower troposphere in January is about 4 to 8 years, but in July 1 to 2 years.**

[Figure]

**Patra et al., 2009, Figure 14. Latitude-pressure distribution of monthly-average instantaneous $CH_4$ lifetime (=1.0 / [$K_{O1D} \times O^1D$ +**
40   **$K_{OH} \times OH + K_{Cl} \times Cl$]) at model grids during (a) boreal winter and (b) boreal summer of 2000.**

The main sink is the oxidation with OH radicals, which is primarily produced by the photolysis of ozone in the presence of water vapor (Saunois et al., 2020). That means, during summer, higher temperature and more sunlight can lead to higher concentration of OH. But other factors like atmospheric circulation impact the lifetime essentially

45 (Patra et al., 2009).

In total, the global lifetime remains the same, but at specific locations and times, the instantaneous lifetime can vary depending on the environmental conditions like concentration of OH radicals, atmospheric circulation pattern etc.

Therefore, knowing the instantaneous lifetime, we cannot simply derive the oxidation fraction of methane per month, because we would need to know the concentration of OH, the presence of other atmospheric gases, environmental

50 conditions at that given month and location etc.

However, Chandra et al., 2021, Fig. 11a, simulated the average monthly removal rate of $CH_4$ over the course of one year. At 30° N, the removal rate is about 60–40 ppb per month.

[Figure]

Fig. 11. Latitude–height distributions of annual (2010) average rate of change in $CH_4$ concentration (tendency) due to the chemical loss (a) and three transport terms (b, c, d: due to advection, convection, and diffusion, respectively) as simulated by the MIROC4-ACTM. The height (y-axis) is shown as mean pressure at model levels, normalized by the surface pressure, as MIROC4-ACTM follows hybrid sigma and pressure coordinate, respectively, below and above 329 hPa or model level 14.

55

Since the oxidation fraction per month for the summer month is not crucial to the understanding of our new approach, we didn't include it in the revised text. However, we clarified the sentence and terminology "instantaneous lifetime" as follows:

Lines 210–214: *During **boreal** summer, **a higher OH concentration contributes to an increased** $CH_4$ **removal** by oxidation*
60 *at our study region (**Travis et al., 2020**). Including other atmospheric factors, such as atmospheric circulation pattern,*
*models estimate the instantaneous lifetime **of** $CH_4$ **for July to be** as short as 1 year (**Fig. 14 in** Patra et al., 2009).*

Figures 3 and 4 are not clear. I suggest to not show the shaded areas, but only the best estimates together with a single bars for the full period that would indicate the typical uncertainty range

65 **Response**

**We revised Fig. 3 and 4 by only showing the 16 ppb uncertainty range of the best result, approach 3 (blended obs. XCH4), and ACTM$_{XCH4}$ as grey area. Furthermore, we removed the comparison with the TCCON stations from Fig. 3 to make the comparison of the approaches clearer. Instead, we added a new Fig. 4 which only shows the results of approach 3 in comparison with those of the two TCCON stations. As pointed out by Referee 3#, the missing legend of**
70 **the linear fit was added to the new Fig. 4.**

**We also revised Fig. 5 (now Fig. 6) and Fig. A3 (now Fig. A4) in order to have the same color depth. In addition, we revised the caption of the new Fig. 6 and new Fig. A4 by adding the description of the uncertainty range:**

Lines 443–445: ***Figure 6:*** *Temporal variation of the blended obs. $XCH_4$ (ACTM$_{XCH4}$, black) in comparison with GOSAT*
75 *$XCH_4$ retrievals from NIES (**orange**), RemoTeC (blue), and OCFP (green) at the latitude range 30–40° N (a) and 20–30° N (b). **The grey area is the 16 ppb uncertainty of the blended obs. $XCH_4$.***

Lines 511–513: ***Figure A4:*** *Temporal variation of the blended obs. $XCH_4$ (ACTM$_{XCH4}$, black) in comparison with GOSAT $XCH_4$ retrievals from NIES (**orange**), RemoTeC Heidelberg (HD) (magenta), RemoTeC SRON (blue), and OCFP (green) at the latitude range g1 = 30–40° N (a) and g2 = 20–30° N (b). **The grey area is the 16 ppb uncertainty range of the blended**
80 **obs. $XCH_4$.***

[Figure]

**Figure 3:** *Temporal variation of monthly averaged XCH₄ obtained by approach 1 (simple obs. XCH₄, green), approach 2 (obs. XCH₄, orange), and approach 3 (blended obs. XCH₄, black) at the latitude range 30–40° N (a) and 20–30° N (b). The uncertainty ranges are 22 ppb, 20 ppb, and 16 ppb for approach 1, 2, and 3 respectively. Only the 16 ppb uncertainty range of approach 3 is shown as grey area. Uncertainty ranges of the other approaches are not shown for readability.*

85

[Figure]

[Figure]

**Figure 4:** *Temporal variation of monthly averaged XCH₄ obtained by approach 3 (blended obs. XCH₄, black), and from the TCCON station in Saga (green) and Tsukuba (orange) at the latitude range 30–40° N (a) and 20–30° N (b). The grey area is the 16 ppb uncertainty range of approach 3; error bars are the standard deviations of TCCON. Also shown is the linear least-square regression (deep blue line) with a 90% confidence interval on the slope and intercept (deep blue dashed line) of approach 3.*

[Figure]

**Figure 5**: *Comparison between the blended obs. XCH₄ (approach 3) derived from CH₄ profiles using the MIROC4-ACTM (ACTM$_{XCH4}$,* **black**)*, CAMS (CAMS$_{XCH4}$,* **green**)*, and CAMSinv (CAMSinv$_{XCH4}$,* **orange***) for the stratospheric column at the latitude range 30–40° N (a) and 20–30° N (b). The uncertainty range **of all results is 16 ppb. The grey area is the uncertainty of ACTM$_{XCH4}$. Uncertainty ranges of the other results are not shown for readability.***

Comment 5

The conclusion is more a summary than a conclusion. It would be better to offer a real conclusion to the reader

**Response**

**Thank you for the comment. It is true that we rather provided a summary of the results of our study than a conclusion. However, we keep the summary part, because we believe, it helps the readers to understand the main results of the study. Based on the summary, we added a real conclusion at the end as shown as response to the following Comment 6.**

**We changed the chapter heading to "5 Summary and Conclusion" to clarify that we give a summary of our results and a conclusion at the end. Beside the main conclusion at the end, we concluded each summary paragraph with one or two sentences as follows:**

Line 446: **5 *Summary and Conclusion***

Lines 461–463: *Based on the* **lowest** *uncertainty and difference towards TCCON, approach 3, defined as blended observation-based XCH₄ (blended obs. XCH₄)*, **is the most suitable for evaluating satellite observations over oceans**.

Lines 468–471: *MIROC4-ACTM and CAMSinv consider chemical losses in the stratosphere,* **where MIROC4-ACTM additionally uses an optimized atmospheric transport model. We conclude that for accurately deriving XCH₄, a well modelled stratosphere is necessary that includes CH₄ sinks. Therefore,** *either CAMSinv or MIROC4-ACTM* **is suitable for our approach** *of which CAMSinv is publicly available.*

Lines 478–479: **These observations show that using the blended obs. XCH₄ dataset, CH₄ trends and seasonal variations can be detected, and satellite observations evaluated.**

**Comment 6**

In addition, the last paragraph is not a conclusion but rather a discussion. Please correct

**Response**

**We revised the last paragraph as follows:**

Lines 480–494: **Having an uncertainty range lower than the mission targets of GOSAT and TROPOMI, the accuracy of satellite derived XCH₄ over oceans can be accessed by our best approach 3. While the blended obs. XCH₄ dataset is not suitable for detecting small scale variations of CH₄ like those from point sources and sinks, spatial pattern and large-scale long-term trends can be evaluated and used for carbon cycle studies. Furthermore, our ship-aircraft based approach has the potential to quickly create long-term dataset in areas where other highly precise reference data, such as from measurement campaigns like HIPPO flights or TCCON stations, are not available. Uncertainties and limitations caused by** limited in situ data **will be reduced** in the near future. **This includes the re-start of** aircraft observations by CONTRAIL over the western Pacific Ocean, probably within the next 2 years, **and the spatial extension** of other aircraft projects like that of the In-service Aircraft for a Global Observing System (IAGOS) project. As a complement to established validation networks we can contribute with our ship-aircraft derived XCH₄ dataset to the validation of TROPOMI, GOSAT-GW and other upcoming satellite missions in future.

**References**

Chandra, N., Patra, P. K., Bisht, J. S. H., Ito, A., Umezawa, T., Saigusa, N., Morimoto, S., Aoki, S., Janssens-Maenhout, G., Fujita, R., Takigawa, M., Watanabe, S., Saitoh, N., and Canadell, J. G.: Emissions from the oil and gas sectors, coal mining and ruminant farming drive methane growth over the past three decades, Journal of the Meteorological Society of Japan, 99, 309–337, https://doi.org/10.2151/jmsj.2021-015, 2021.

K. Patra, P., Takigawa, M., Ishijima, K., Choi, B.-C., Cunnold, D., J. Dlugokencky, E., Fraser, P., J. Gomez-Pelaez, A., Goo,
T.-Y., Kim, J.-S., Krummel, P., Langenfelds, R., Meinhardt, F., Mukai, H., O'Doherty, S., G. Prinn, R., Simmonds, P.,
Steele, P., Tohjima, Y., Tsuboi, K., Uhse, K., Weiss, R., Worthy, D., and Nakazawa, T.: Growth Rate, Seasonal,
Synoptic, Diurnal Variations and Budget of Methane in the Lower Atmosphere, Journal of the Meteorological Society
of Japan. Ser. II, 87, 635–663, https://doi.org/10.2151/jmsj.87.635, 2009.

Saunois, M., Stavert, A., Poulter, B., Bousquet, P., Canadell, J., Jackson, R., Raymond, P., Dlugokencky, E., Houweling, S.,
Patra, P., Ciais, P., Arora, V., Bastviken, D., Bergamaschi, P., Blake, D., Brailsford, G., Bruhwiler, L., Carlson, K.,
Carrol, M., Castaldi, S., Chandra, N., Crevoisier, C., Crill, P., Covey, K., Curry, C., Etiope, G., Frankenberg, C.,
Gedney, N., Hegglin, M., Höglund-Isaksson, L., Hugelius, G., Ishizawa, M., Ito, A., Janssens-Maenhout, G., Jensen,
K., Joos, F., Kleinen, T., Krummel, P., Langenfelds, R., Laruelle, G., Liu, L., Machida, T., Maksyutov, S., McDonald,
K., McNorton, J., Miller, P., Melton, J., Morino, I., Müller, J., Murguia-Flores, F., Naik, V., Niwa, Y., Noce, S.,
O'Doherty, S., Parker, R., Peng, C., Peng, S., Peters, G., Prigent, C., Prinn, R., Ramonet, M., Regnier, P., Riley, W.,
Rosentreter, J., Segers, A., Simpson, I., Shi, H., Smith, S., Steele, L. P., Thornton, B., Tian, H., Tohjima, Y., Tubiello,
F., Tsuruta, A., Viovy, N., Voulgarakis, A., Weber, T., van Weele, M., van der Werf, G., Weiss, R., Worthy, D.,
Wunch, D., Yin, Y., Yoshida, Y., Zhang, W., Zhang, Z., Zhao, Y., Zheng, B., Zhu, Q., Zhu, Q., and Zhuang, Q.: The
Global Methane Budget 2000–2017, Earth Syst Sci Data, 12, 1561–1623, https://doi.org/10.5194/essd-12-1561-2020,
2020.

Szopa, S., Naik, V., Adhikary, B., Artaxo, P., Berntsen, T., Collins, W. D., Fuzzi, S., Gallardo, L., Kiendler Scharr, A.,
Klimont, Z., Liao, H., Unger, N., and Zanis, P.: Short-Lived Climate Forcers, 817–922 pp.,
https://doi.org/10.1017/9781009157896.008, 2021.

Travis, K. R., Heald, C. L., Allen, H. M., Apel, E. C., Arnold, S. R., Blake, D. R., Brune, W. H., Chen, X., Commane, R.,
Crounse, J. D., Daube, B. C., Diskin, G. S., Elkins, J. W., Evans, M. J., Hall, S. R., Hintsa, E. J., Hornbrook, R. S.,
Kasibhatla, P. S., Kim, M. J., Luo, G., McKain, K., Millet, D. B., Moore, F. L., Peischl, J., Ryerson, T. B., Sherwen, T.,
Thames, A. B., Ullmann, K., Wang, X., Wennberg, P. O., Wolfe, G. M., and Yu, F.: Constraining remote oxidation
capacity with ATom observations, Atmos Chem Phys, 20, 7753–7781, https://doi.org/10.5194/acp-20-7753-2020,
2020.

---

## Author Comment (AC2)

**General.**

We would like to thank the anonymous Referee #2 for providing very valuable comments to improve and clarify our manuscript. Many of the questions regarding the uncertainty calculation are related. Therefore, our responses to various questions contain cross-references. Please find our responses to the specific comments and questions below. Our response is
5    written in bold. The revised parts of the manuscript are highlighted in bold red.

**Comments of Referee #2 and our responses to them**

**General comments**

The authors develop approaches for generating XCH4 time series over ocean combining ship and aircraft measurements with model data. The observation-based XH4 data are compared with independent TCCON measurements and finally used for
10    evaluating GOSAT measurements. The paper is well written and within scope of AMT.

I have some minor comments related to the uncertainty calculation that should be addressed in a revised manuscript:

**Specific comments**

**Comment 1**

L82: What would be the required accuracy for a dataset assess the accuracy of trends and variations in XCH4 satellite
15    observations over oceans? Moreover, what is the accuracy that is achieved with the dataset presented in this study?

**Response**

**In order that our dataset is useful for accessing the accuracy of trends and variations in satellite data, the uncertainty of our reference dataset should be lower than that of the satellites.**

**For GOSAT, launched in 2009, the target for CH$_4$ was a relative accuracy of 2% for 3-month averaged data within a**
20    **1,000$^2$ km$^2$ grid. The target was achieved in 2010. This accuracy is suitable for research on global phenomena and for getting a better understanding of carbon cycles (Nakajima et al., 2010).**

**The mission targets for TROPOMI for the total column of CH$_4$ are a systematic error (bias) of less than 1.5% and 1% precision (ESA, 2017). Because the accuracy is determined by both, the bias and precision, it would be in the range of 1.8% using Gaussian Error propagation. This corresponds to concentrations of around 30 ppb.**

25 **Higher accuracy is required for the estimation of regional sources and sinks, for example for political decision making related to global warming countermeasures. Thresholds are given for land observations, and they are much higher with a precision of < 34 ppb for a single observation and < 11 ppb for monthly averaged data within $1000^2$ km$^2$ grid. The systematic error after bias correction should be < 10 ppb (Buchwitz et., al, 2020).**

**In this context, GOSAT-2 was launched in 2018 with the aim for improved concentration precision of 5 ppb for**
30 **monthly averaged CH$_4$ data at $500^2$ km$^2$ grid over land and $2000^2$ km$^2$ grid over the ocean (Nakajima et al., 2017).**

**Given the above mission targets of GOSAT and TROPOMI, our dataset with a conservative estimated uncertainty of 16 ppb fulfils the requirement.**

**Besides the uncertainty, the long-term availability of a reference dataset is important in areas where no other long-term datasets are available. Therefore, in regions like the open ocean, a reference dataset with even a high**
35 **uncertainty is useful to fill in gaps where other highly precise reference data, such as from measurement campaigns like HIPPO flights or TCCON stations, are not available. Even though the reference has a relative high uncertainty, spatial pattern and large-scale long-term trends can be evaluated. Our dataset is not suitable for detecting small scale variations like those from point sources and sinks.**

40 **We clarified the requirement as follows:**

Lines 83–87: *We propose a new approach to assess the accuracy of satellite derived XCH$_4$ trends and variations over open ocean regions by combining commercial ship and various aircraft observations with the help of atmospheric chemistry models. **We are targeting an accuracy better than that required for the GOSAT and TROPOMI mission of <35 ppb (<2%) (ESA, 2017; Nakajima et al., 2010). Our** approach was successfully applied to the evaluation of satellite XCO$_2$ previously*
45 *(Müller et al., 2021).*

Lines 621–623: ***ESA, European Space Agency: Sentinel-5 Precursor Calibration and Validation Plan for the Operational Phase, Issue 1, Revision 1, 26 pp., https://sentinel.esa.int/documents/247904/2474724/Sentinel-5P-Calibration-and-Validation-Plan.pdf, 2017, accessed on 28 November 2023.***

Lines 690–692: ***Nakajima, M., Kuze, A., Kawakami, S., Shiomi, K., and Suto, H.: Monitoring of the greenhouse gases***
50 ***from space by GOSAT, International Archives of the Photogrammetry, Remote Sensing and Spatial Information Sciences - ISPRS Archives, 38, 94–99, 2010.***

Comment 2

Figure 1 could already be mentioned in the beginning of Section 2.

**Response**

We added references to Figure 1 in section 2 as follows:

Lines 93–95: *As part of Japan's Comprehensive Observation Network for Trace gases by Airliner, CONTRAIL, air samples of $CH_4$ are collected by the Automatic air Sampling Equipment (ASE) and Manual air Sampling Equipment (MSE) about twice a month between Japan, Hawaii, and Australia since 2005.* **The sampling locations of the CONTRAIL data are shown in Fig. 1.**

Lines 122–124: *In this study, we used $CH_4$ observations by the cargo ship Trans Future 5 (TF5, Toyofuji Shipping Co., Ltd.), which sails between Japan, Australia, and New Zealand* **(Fig. 1)**.

Comment 3

Section 3.3.1: The calculation of the tropospheric uncertainty of XCH4 is difficult to judge mainly because no profiles are shown in the manuscript. Please add a figure comparing the constructed CH4 profiles from measurements and the MIROC4-ACTM model with the HIPPO profiles.

**Response**

**Thank you for the comment.**

**Below, we show the comparison between MIROC4-ACTM and HIPPO 4 profiles under a) in Fig. 2 and Table 1.**

**The comparison between MIROC4-ACTM and obs. $CH_4$ profiles is shown under b) in Fig. 3 using some example profiles, and Table 2.**

**However, we cannot show the direct comparison between HIPPO 4 profiles and the constructed obs. $CH_4$ profiles. You can find our explanation under the response to Comment 4.**

**For the comparison, we selected 8 HIPPO profiles, which are within 2000 km distance of the centre location of the bounding box g2. See Fig. 1 below. We like to clarify that in the original manuscript mistakenly only 6 profiles were selected. This is corrected in the revised manuscript, and changes resulting from this are listed under our response to Comment 6.**

**More details about the reason for the selection of these 8 profiles are found under our response to Comment 5.**

**a) MIROC4-ACTM versus HIPPO 4**

**Figure 1 below shows the location of the profiles within the 2000 km buffer, and Fig. 2 the HIPPO 4 profiles in comparison with those for the MIROC4-ACTM on July 3 and 6, 2011, respectively. We added Fig. 2 to Appendix A**

of the manuscript as Fig. A1. The average difference ± standard deviation of the differences and root-mean-square error (RMSE) between the profiles were 6 ± 5 ppb (RMSE = 8), 6 ± 10 ppb (RMSE = 12), 6 ± 12 ppb (RMSE = 13) for the altitude ranges 0-1500 m, 1500-6000 m, and 6000-11000m, respectively (Table 1).

85

[Figure]

**Figure 1.** Location of selected HIPPO 4 profiles on July 3 and 6, 2011 within 2000 km distance of the centre location of bounding box g2.

[Figure]

90   **Figure 2.** Comparison between HIPPO 4 (blue) and MIROC4-ACTM profiles (red) on July 3 (a) and 6 (b), 2011.

**Table 1.** Average difference between MIROC4-ACTM (ACTM) and HIPPO 4 data (mean difference ± standard deviation of differences) and root-mean-square error (RMSE) at different altitude ranges.

| Altitude [m] | ACTM – HIPPO 4 [ppb] | RMSE [ppb] |
|---|---|---|
| **0-1500** | 6 ± 5 | 8 |
| **1500-6000** | 6 ± 10 | 12 |
| **6000-11000** | 6 ± 12 | 13 |

95

**b) MIROC4-ACTM versus obs. CH4 profiles**

**Figure 3 shows the MIROC4-ACTM (black) and the constructed obs. CH₄ profiles. For illustration, only some examples are shown. Table 2 lists the mean difference ± standard deviation of differences and the RMSE for each altitude range.**

100

[Figure]

**Figure 3.** Comparison MIROC4-ACTM (MIROC4 prof) with obs. CH₄ profiles. The profiles are examples. The red curve is the MIROC4-ACTM profile interpolated on the pressure grid of the obs. CH₄ profile.

**Table 2.** Average difference between MIROC4-ACTM (ACTM) and obs. $CH_4$ profile data (mean difference ± standard deviation of differences) and root-mean-square error (RMSE) at different altitude ranges

| Altitude [m] | ACTM – obs. $CH_4$ | RMSE [ppb] |
|---|---|---|
| 0-1500 | 4 ± 18 | 18 |
| 1500-6000 | −3 ± 17 | 17 |
| 6000-11000 | 5 ± 16 | 18 |

Comment 4

It would also be interesting to see how well your approaches can reconstruct a HIPPO profile when taking the three measurements (2 aircraft + 1 ship) from the HIPPO profile.

**Response**

**We agree with the referee #2 that this comparison would be very interesting and important to evaluate our approach. However, there are several reasons why we cannot provide a reasonable comparison.**

**The reasons are as follows.:**

- **The HIPPO profiles are obtained only on 2 days of July 2011 (July 3 and 6) at specific locations.**
- **In contrast, our approach is based on monthly averaged data within a 10° latitude by 20° longitude grid. If we increase the sampling frequency to, for example, ± 2 days within 1 degree of the HIPPO profiles or higher, we won't have enough in situ data to apply our approach.**
- **Furthermore, our current data processing was for the years 2014–2017 for the latitude range north of the HIPPO flights. One reason for selecting that location was that we have additional JMA data for ship and aircraft. These data are missing at the location of the HIPPO flights, which makes the number of in situ data even less.**
- **If we apply our approach of monthly averages using the 10° latitude by 20° longitude grid for July 2011, we will not be able to reproduce the strong variation of a specific HIPPO profile of a single flight or day.**

**However, if in future more in situ data are available, and new profile flights are performed, we agree that this comparison is very important!**

Comment 5

L249: Can you explain why you only used only six profiles for assessing the MIROC4-ACTM simulations, while 20 profiles seem to be available in the study region?

**Response**

First, we want to clarify that 8 HIPPO profiles should have been included and not 6, which was a mistake in our calculation previously. Second, we noticed that the root-mean-square error (RMSE), which is as a measure of the differences between the model and the in situ observations, is a better and straight forward measure to access the uncertainty instead of using the average ± standard deviation of the differences. Therefore, the uncertainty numbers changed in the revised manuscript. The changes made are listed under the response to Comment 6.

Our study region is influenced by the continental emission outflow. These conditions are expected to be more challenging to be represented by the model correctly. To assess the uncertainty of the MIROC4-ACTM for conditions similar to our study region, we selected HIPPO flights within 2000 km of the centre location of the grid box g2. We chose the 2000 km threshold as a balance between closeness of the profile flights and number (Fig. 1).

Comment 6

L255: Finally, it is unclear how the uncertainty in the profiles translate to XCH4 uncertainties. Do you do an error propagation or sensitivity study as for the tropopause uncertainty?

**Response**

Using the 8 selected HIPPO 4 profiles as a reference, we can only access the tropospheric uncertainty of our constructed profiles indirectly. We used Gaussian Error Propagation as described as follows:

**1)**

First, we assess the uncertainty of the MIROC4-ACTM profiles by calculating the difference between MIROC4-ACTM and HIPPO 4 to derive the RMSE as measure for the uncertainty of the model. Here we call it "ACTM_unc".

**2)**

Second, we estimate the uncertainty of our obs. $CH_4$ profile in 2 steps:

a) We calculate the difference between obs. $CH_4$ profile and MIROC4-ACTM profile and obtain the RMSE as part of the total tropospheric uncertainty of the obs. $CH_4$ profile.

b) The total uncertainty consists of the partial uncertainty a) + that of the ACTM model (ACTM_unc) from step 1. It can be calculated using Gaussian Error Propagation. The results are shown in Table 3.

**Table 3.** Root-mean-square error of the difference between MIROC4-ACTM (ACTM) and HIPPO 4, and MIROC4-ACTM and obs. $CH_4$ profile data at different altitude ranges. Last column shows the total uncertainty after Gaussian Error propagation. Uncertainties applied to approach 3 are shown in bold.

| Altitude [m] | ACTM – HIPPO 4 | ACTM – obs. $CH_4$ | Total uncertainty |
|---|---|---|---|
| **0-1500** | 8 | 18 | **20** |
| **1500-6000** | 12 | 17 | **21** |
| **6000-11000** | **13** | 18 | 22 |

For approach 3, MIROC4-ACTM data are used at the altitude range 6000-11000 m. Therefore, no error propagation was applied and only the ACTM_unc was used for that altitude range (RMSE = 13). Uncertainties of approach 3 are shown bold in Table 3.

**We revised Table 1 in the manuscript as follows:**

**Table 1:** Uncertainty assessment of the obs. $CH_4$ profiles at the troposphere. Top rows: average concentration range of $CH_4$ within each HIPPO 4 profile (mean variability ± standard deviation). Bottom rows: **Root-mean-square error (RMSE) of the difference between MIROC4-ACTM (ACTM) and HIPPO 4, and MIROC4-ACTM and obs. $CH_4$ profile data at different altitude ranges. The last column shows the total uncertainty after Gaussian Error propagation. Uncertainties applied to approach 3 are shown in bold.**

| HIPPO 4 profile range [m] | Variation within profiles [ppb] | | |
|---|---|---|---|
| ~300–~13000 | 24 ± 17 | | |

| Altitude [m] | ACTM – HIPPO 4 [ppb] | ACTM – obs. $CH_4$ [ppb] | Total uncertainty [ppb] |
|---|---|---|---|
| 0–1500 | 8 | 18 | 20 |
| 1500–6000 | 12 | 17 | **21** |
| 6000–11000 | **13** | 18 | 22 |

**A detailed description of our uncertainty estimation in the troposphere is added as follows:**

Lines 260–274: *Second, we assessed **the uncertainty of the constructed $CH_4$ profiles in 3 steps with the help of the MIROC4-ACTM. In the first step, we investigate** how good the MIROC4-ACTM reproduces the variation of HIPPO profiles **for similar conditions to our study region, which is influenced by the continental emission outflow (Appendix A, Fig. A1). Therefore, we selected 8 profiles within 2000 km of the center location of g2** (**Fig. 1**). We choose the MIROC4-ACTM to be consistent with our previous study (Müller et al., 2021). We distinguished the altitude range 0–1500 m, corresponding to the boundary layer, 1500–6000 m, corresponding to the middle troposphere between the extrapolated ship and JMA aircraft data, and 6000–11000 m, corresponding to the upper troposphere between the JMA and CONTRAIL aircraft data. **As model uncertainty, we obtain the root-mean-square error (RMSE) of the** difference between the MIROC4-ACTM and the HIPPO profiles **with 8** ppb, **12** ppb, and **13** ppb for the altitude ranges 0–1500 m, 1500–6000 m, and 6000–11000 m, respectively (**Table 1**). **In the second step, we compare the MIROC4-ACTM with our obs. $CH_4$ profiles and obtain the RMSE (Table 1, ACTM – obs. $CH_4$). Because the model itself has an uncertainty as obtained in step 1, the tropospheric uncertainty of the constructed profile of each altitude range is 20 ppb, 21 ppb, and 22 ppb using Gaussian Error propagation (Table 1, Total uncertainty). As a result**, we added **21** ppb uncertainty between the extrapolated ship and JMA data **in approach 2 and 3**, and **22** ppb **and 13 ppb** between the JMA data and up to the TROPPB in approach 2 and 3, **respectively***.

**The updated uncertainty values are added as follows:**

Line 23: *Uncertainties were 22 ppb for approach 1, **20 ppb** for approach 2**, and **16 ppb for approach** 3.*

Lines 328–329: *The uncertainty range of the simple obs. XCH$_4$ (22 ppb) is by **2 and 6** ppb larger than those of the obs. XCH$_4$* **(20 ppb)** *and blended obs. XCH$_4$ (1**6** ppb),* **respectively** *(**section 3.3**).*

Lines 365–367: *Given the lower maximal possible averaged difference between TCCON and approach 2 and 3 compared to approach 1,* **and given the lowest uncertainty range of approach 3,** *the latter approach* **is** *preferable for future applications.*

Lines 430–431: *The retrievals mostly lie in the uncertainty range (1**6** ppb) of the blended obs. XCH$_4$.*

Lines 454–455: *Uncertainties of the calculated XCH$_4$ were reduced by **2** ppb **and 6 ppb** from 22 ppb (approach 1) to **20** ppb for approach 2 and **16 ppb for approach 3.***

**The Figure numbers of the Appendix changed as follows:**

Lines 180–182: *A comparison with the RemoTeC v2.4.0 full-physics retrieval operated at Heidelberg University is shown in* **Appendix A (Fig. A4).**

Lines 296–297: *GOSAT NIES CH$_4$ observations have a higher sensitivity in the stratospheric column as compared to CO$_2$ (averaging kernel >0.8 in the stratosphere,* **Appendix A, Fig. A2**).

Lines 308–309: *CAMS was positively biased by 138 ± 9 ppb, and 165 ± 15 ppb at 30–40° N and 20–30° N, respectively* (**Table 2, Appendix A, Fig. A3 (a), (b)**).

Lines 311–313: *The highest average difference occurred in June (30–40° N: 37 ± 6 ppb, 20–30° N: 44 ± 3 ppb), the lowest in October (4 ± 0.6 ppb) at 30–40° N, and January (5 ± 5 ppb) and February (3 ± 13 ppb) at 20–30° N (**Appendix A, Fig. A3 (c), (d)**).*

**Furthermore, Table S5 and S6 of the supplement are updated with the new uncertainties of the obs. XCH$_4$ of 20 ppb, and the blended obs. XCH$_4$ of 16 ppb.**

Comment 7

L337: If you have three simulations and two agree with each other, it is not valid to conclude that the agreeing models are correct. The conclusion in this paragraph need therefore to be argued more carefully using previous results from literature (as done in Section 3.3.3) or conducting additional analyses (e.g., comparison with independent measurements).

**Response**

**We agree with the referee that our argumentation needs to be more careful.**
**We added the clarification as follows:**

Lines 370–380: ***Figure 5*** *shows the comparison of the blended obs. XCH₄ (approach 3) using the MIROC4-ACTM, CAMS, and CAMSinv for the stratospheric column (**section 3.3.3**), denoted as ACTMₓCH₄, CAMSₓCH₄, and CAMSinvₓCH₄. Using ACTMₓCH₄ as reference, CAMSₓCH₄ is highly biased at both latitude ranges by 12 ± 5 ppb (0.6 ± 0.2%) in total. In contrast, CAMSinvₓCH₄ shows a small negative total bias of −5 ± 3 ppb (−0.3 ± 0.2%).* ***CAMS has a known large positive***

**stratospheric CH₄ bias (Agustí-Panareda et al., 2023). MIROC4-ACTM and CAMSinv account for stratospheric CH₄ loss and the modelled stratosphere is comparable as discussed in section 3.3.3.** *The similarity of* ***the ACTMₓCH₄ and CAMSinvₓCH₄*** *and their differences to CAMSₓCH₄* ***indicate*** *the strong impact of the stratospheric part on the derived XCH₄ and highlights the importance to make an appropriate model choice.* ***Considering the large uncertainty of CAMS and the fact that the other two products are better optimized for modelling CH₄ in the stratosphere,*** *we suggest using either the*

**MIROC4-ACTM or CAMSinv to model the stratospheric column.**

**Furthermore, we made the following correction in section 3.3.3. CAMSinv is "inversion-optimized for greenhouse fluxes and concentrations" but doesn't use a better atmospheric transport model than CAMS. We revised the sentence as follows:**

Lines 315–317: *Compared to CAMS, both the MIROC4-ACTM and CAMSinv account for chemical losses in the stratosphere.* ***Additionally, MIROC4-ACTM uses an optimized atmospheric transport model (Patra et al., 2018).***

Lines 468–471: *MIROC4-ACTM and CAMSinv consider chemical losses in the stratosphere,* ***where MIROC4-ACTM additionally uses an optimized atmospheric transport model. We conclude that for accurately deriving XCH₄, a well modelled stratosphere is necessary that includes CH₄ sinks. Therefore,*** *either CAMSinv or MIROC4-ACTM is suitable for our approach of which CAMSinv is publicly available.*

**Reference**

Agustí-Panareda, A., Barré, J., Massart, S., Inness, A., Aben, I., Ades, M., Baier, B. C., Balsamo, G., Borsdorff, T., Bousserez, N., Boussetta, S., Buchwitz, M., Cantarello, L., Crevoisier, C., Engelen, R., Eskes, H., Flemming, J., Garrigues, S., Hasekamp, O., Huijnen, V., Jones, L., Kipling, Z., Langerock, B., McNorton, J., Meilhac, N., Noël, S., Parrington, M., Peuch, V.-H., Ramonet, M., Razinger, M., Reuter, M., Ribas, R., Suttie, M., Sweeney, C., Tarniewicz, J., and Wu, L.: Technical note: The CAMS greenhouse gas reanalysis from 2003 to 2020, Atmos Chem Phys, 23, 3829–3859, https://doi.org/10.5194/acp-23-3829-2023, 2023.

Buchwitz, M., Chevallier, F., and Marshall, J.: User Requirements Document (URD) - Greenhouse Gases (GHG), Simulation, 42, 2020.

Müller, A., Tanimoto, H., Sugita, T., Machida, T., Nakaoka, S., Patra, P. K., Laughner, J., and Crisp, D.: New approach to evaluate satellite-derived XCO2 over oceans by integrating ship and aircraft observations, Atmos. Chem. Phys., 21, 8255–8271, https://doi.org/10.5194/acp-21-8255-2021, 2021.

Nakajima, M., Kuze, A., Kawakami, S., Shiomi, K., and Suto, H.: Monitoring of the greenhouse gases from space by GOSAT, International Archives of the Photogrammetry, Remote Sensing and Spatial Information Sciences - ISPRS Archives, 38, 94–99, 2010.

Nakajima, M., Suto, H., Yotsumoto, K., Shiomi, K., and Hirabayashi, T.: Fourier transform spectrometer on GOSAT and GOSAT-2, in: International Conference on Space Optics — ICSO 2014, 2, https://doi.org/10.1117/12.2304062, 2017.

Patra, P. K., Takigawa, M., Watanabe, S., Chandra, N., Ishijima, K., and Yamashita, Y.: Improved Chemical Tracer Simulation by MIROC4.0-based Atmospheric Chemistry-Transport Model (MIROC4-ACTM), SOLA, 14, 91–96, https://doi.org/10.2151/sola.2018-016, 2018.

ESA, European Space Agency: Sentinel-5 Precursor Calibration and Validation Plan for the Operational Phase, Issue 1, Revision 1, 26 pp., https://sentinel.esa.int/documents/247904/2474724/Sentinel-5P-Calibration-and-Validation-Plan.pdf, 2017, accessed on 28 November 2023.

---

## Author Comment (AC3)

**General.**

We would like to thank the anonymous Referee #3 for providing comments to improve and clarify our manuscript. We will revise the text by fully taking the comments into account. Please find our responses to the specific comments and questions below. Our response is written in bold. The revised parts of the manuscript are highlighted in bold red.

**Comments of Referee #3 and our responses to them**

**General comments**

Muller et al., developed three approaches to construct the CH4 vertical profiles, and then use them to calculate the XCH4. They then show how these observation-based XCH4s can be used to investigate the seasonal variation in CH4 and to evaluate the satellite observations of CH4. This study involves a large amount of data, including airborne and ship 10 measurements, satellite observations, and model simulations. This manuscript is well-organized and is within the scope of AMT. I recommend its publication after the authors address the following comments:

**Specific comments**

Comment 1

Sampling bias is a big concern as the ship observations are 6 +- 4 days and the airborne observations are 2+-1 days each 15 month. The uncertainty arising from limited measurements is not well covered in Sect 3.3.1.

**Response**

**Thank you for pointing this out. The uncertainty due to the limited number of in situ data is currently the major drawback of our approach. We added the following explanation at the beginning of chapter 3.3:**

Lines 241–246: *There are two uncertainty sources. The first uncertainty source arises from the limited number and spatiotemporal distribution of in situ data within the latitude-longitude boxes of each month. Therefore, the data may not always represent the monthly averaged $CH_4$ concentration within the area of interest accurately. However, in the near future, the number of in situ data will increase and the spatial distribution expands as discussed in chapter 5. The second source of uncertainties in the obs. $XCH_4$ (simple, blended) are caused by the $CH_4$ profile construction: a) the inter- and* 25 *extrapolation of the in situ data in the troposphere, b) the tropopause height, and c) the modelled stratospheric column.*

Comment 2

Figure 2 includes a lot of information. I would add observation-based profiles from 3 approaches in different colors, instead of showing them in one symbol.

**Response**

**We agree that Fig. 2 was not clear. The original Fig. 2 showed the interpolated observation-based profile of approach 2. The other two approaches were not specifically shown. Therefore, we revised Fig. 2 by adding subfigures for each approach and added references to each subfigure to the text as follows below.**

[Figure]

**Figure 2:** Construction of the observation-based $CH_4$ profile (blue) obtained by using ship and aircraft data (yellow) together with model results (green), and the interpolation onto the pressure grid of the satellite retrieval (red) **for approach 1 a), approach 2 b), and approach 3 c)**. The example is obtained at the latitude 30–40° N, in **March** 2015.

Line 223: *Approach 1 is the adaptation of the approach of Müller et al. (2021)* *(Fig. 2a)*.

Lines 226–228: *Approach 2 is the addition of JMA aircraft data to the mid troposphere* *(Fig. 2b)*. *We linearly interpolate between the extrapolated ship data, and both aircraft data. In approach 3, we fill in model results between the aircraft data of JMA and CONTRAIL of approach 2* *(Fig. 2c)*.

Figures 3 and 4 are very difficult to read. It is also unclear where is the gap in the dataset. In Figure 3, the linear regression fitted line is not in the legend and has the same color as blended obs/ XCH4.

**Response**

**In accordance with a similar comment of referee #1, we revised Fig. 3 and 4 by only showing the 16 ppb uncertainty range of approach 3 (blended obs. XCH₄), and ACTM_XCH4 as grey area. Furthermore, we removed the comparison**

**with the TCCON stations from Fig. 3 to make the comparison of the approaches clearer. Instead, we added a new Fig. 4 which only shows results of approach 3 in comparison with those of the two TCCON stations. The linear fit is added to the legend in a color different to that of the blended XCH₄.**

**Data points are connected by straight lines. In the revised Figures with reduced information, the data gabs become clearer, seen as long straight lines between the markers.**

**We also revised Fig. 5 (now Fig. 6) and Fig. A3 (now Fig. A4) in order to have the same color depth. In addition, we revised the caption of the new Fig. 6 and new Fig. A4 by adding the description of the uncertainty range:**

Lines 443–445: *Figure 6: Temporal variation of the blended obs. XCH₄ (ACTM_XCH4, black) in comparison with GOSAT XCH₄ retrievals*

*from NIES (orange), RemoTeC (blue), and OCFP (green) at the latitude range 30–40° N (a) and 20–30° N (b). **The grey area is the 16 ppb uncertainty of the blended obs. XCH₄.***

Lines 511–513: *Figure A4: Temporal variation of the blended obs. XCH₄ (ACTM_XCH4, black) in comparison with GOSAT XCH₄ retrievals from NIES (orange), RemoTeC Heidelberg (HD) (magenta), RemoTeC SRON (blue), and OCFP (green) at the latitude range g1 = 30–40° N (a) and g2 = 20–30° N (b). **The grey area is the 16 ppb uncertainty range of the blended***

***obs. XCH₄.***

[Figure]

**Figure 3:** Temporal variation of monthly averaged XCH$_4$ obtained by approach 1 (simple obs. XCH$_4$, green), approach 2 (obs. XCH$_4$, orange), and approach 3 (blended obs. XCH$_4$, black) at the latitude range 30–40° N (a) and 20–30° N (b). The uncertainty ranges are 22 ppb, 20 ppb, and 16 ppb for approach 1, 2, and 3 respectively. Only the 16 ppb uncertainty range of approach 3 is shown as grey area. Uncertainty ranges of the other approaches are not shown for readability.

[Figure]

[Figure]

**Figure 4:** Temporal variation of monthly averaged XCH₄ obtained by approach 3 (blended obs. XCH₄, black), and from the TCCON station in Saga (green) and Tsukuba (**orange**) at the latitude range 30–40° N (a) and 20–30° N (b). **The grey** area **is** the **16 ppb** uncertainty range of approach **3**; error bars are the standard deviations of TCCON. Also shown is the linear least-square regression (**deep blue** line) with a 90% confidence interval on the slope and intercept (**deep blue** dashed line) of approach 3.

[Figure]

**Figure 5:** Comparison between the blended obs. XCH4 (approach 3) derived from CH4 profiles using the MIROC4-ACTM (ACTM$_{XCH4}$, **black**), CAMS (CAMS$_{XCH4}$, **green**), and CAMSinv (CAMSinv$_{XCH4}$, **orange**) for the stratospheric column at the latitude range 30–40° N (a) and 20–30° N (b). **The uncertainty range of all results is 16 ppb. The grey area is uncertainty of ACTM$_{XCH4}$. Uncertainty ranges of the other results are not shown for readability.**

Comment 4

Line 154: Why do you choose the data that only assimilates NOAA surface observation? How is it different from assimilation using both NOAA surface observations and GOSAT observations?

**Response**

**Our aim is to provide a reference dataset for satellite validation as complement to other networks like TCCON. The model used in our approach has to be independent from the satellite retrieval, which we want to evaluate. Therefore, we do not use data which assimilates GOSAT data. However, the assimilation of precise in situ data improves the accuracy of model calculations.**

**We clarified the sentence as follows:**

Lines 155–156: *We choose datasets which assimilate NOAA surface observations, but not GOSAT observations **to ensure that the model results in our approach are independent from the satellite we aim to validate**.*

Line 367-370: Why the increasing trend in XCH4 is larger between 20-30N than 30-40N.

**Response**

**The first thing to note is that fewer data at 30-40° N as compared to 20-30° N might have cause an artificial difference in the growth rate between the latitude ranges as mentioned in lines 411–412.**

**Besides this, the growth rate is influenced by the complex interaction of various factors, especially by anthropogenic $CH_4$ emissions, the availability of OH radicals as a primary oxidant for methane, and atmospheric circulation patterns. To confirm that there is a real difference in the growth rate of the neighbouring latitude ranges, more comprehensive analysis is needed.**

**However, given a lower growth rate in the northern latitude range, combined with a higher similarity of our obs. $XCH_4$ with those $XCH_4$ influenced by the Asian emission outflow at Saga (chapter 4.1), we can suggest that the**
**interaction between various anthropogenic emissions might have led to higher OH concentrations, consequently, $CH_4$ removal rates near to the Japanese East coast (Fig. 1), and therefore causing a slower annual growth rate. Another explanation can be the decreasing trend of $CH_4$ emissions from Japan related to policy changes (Ito et al., 2023).**

**We added the following explanation:**

Lines 411–417: It is noted that limited and uneven sampled in situ data during each month might cause an artificial difference **of the growth rates** between the latitude ranges. *However, given a lower growth rate at the higher latitude range combined with a higher similarity of the blended obs. $XCH_4$ with those $XCH_4$ influenced by the Asian emission outflow at Saga (chapter 4.1), we can suggest that the interaction between anthropogenic emissions might have led to increased OH concentrations, consequently higher $CH_4$ removal rates near to the Japanese East coast (Fig. 1), and*
*therefore causing a slower annual growth. Or, it might indicate that, compared to 20–30° N, the higher latitude range is affected by the decreasing trend in $CH_4$ emission from Japan (Ito et al., 2023).*

Lines 643–644: *Ito, A., Patra, P. K., and Umezawa, T.: Bottom-Up Evaluation of the Methane Budget in Asia and Its Subregions, Global Biogeochemical Cycles, 37, https://doi.org/10.1029/2023gb007723, 2023.*

**Other corrections made**

Lines 306–308: *For that, we interpolated the MIROC4-ACTM data with its higher resolved pressure grid on that of the CAMS and CAMSinv data, respectively (section 2.3).*

Lines 538–539: *CAMSinv data were provided by CWO.*

CWO was added under Author contribution beside being not a co-author. We corrected this.

Lines 608–610: *Copernicus Climate Change Service, Climate Data Store: Methane data from 2002 to present derived from satellite observations, Copernicus Climate Change Service (C3S) Climate Data Store (CDS), https://doi.org/10.24381/cds.b25419f8, 2018, accessed on 17 May 2023.*

Lines 695–697: *NIES GOSAT Project: Release Note of Bias-corrected FTS SWIR Level 2 $CO_2$, $CH_4$ Products*
*(V02.95/V02.96) for General Users", https://data2.gosat.nies.go.jp/doc/documents/ReleaseNote_FTSSWIRL2_BiasCorr_V02.95-V02.96_en.pdf, 2020, revised 2021, accessed on 21 April 2023.*

Lines 812–815: *Yoshida, Y., Someya, Y., Ohyama, H., Morino, I., Matsunaga, T., Deutscher, N. M., Griffith, D. W. T., Hase, F., Iraci, L. T., Kivi, R., Notholt, J., Pollard, D. F., Té, Y., Velazco, V. A., Wunch, D.: Quality evaluation of the column-*
*averaged dry air mole fractions of carbon dioxide and methane observed by GOSAT and GOSAT-2, Scientific Online Letters on the Atmosphere (SOLA), 19, 173–184, https://doi.org/10.2151/sola.2023-023, 2023.*

**References**

Ito, A., Patra, P. K., and Umezawa, T.: Bottom-Up Evaluation of the Methane Budget in Asia and Its Subregions, Global
Biogeochemical Cycles, 37, https://doi.org/10.1029/2023gb007723, 2023.